# The pUL47 tegument protein of Marek's Disease Virus interacts with p32/C1QBP to promote horizontal transmission

Mallorie Durand[1☉], Aurélien Chuard[1☉], Sylvie Rémy[1], Katia Courvoisier-Guyader[1], Sébastien Leclercq[2], Julien Pichon[3], Caroline Denesvre[1], David Pasdeloup[1]*

**1** Laboratory of Biology of Avian Viruses, INRAE-Université de Tours, UMR1282 ISP, Nouzilly, France, **2** Laboratory of Genomic Plasticity, Biodiversity, and Antimicrobial Resistance, INRAE-Université de Tours, UMR1282 ISP, Nouzilly, France, **3** Laboratory of Imaging and Infectiology, INRAE-Université de Tours, UMR1282 ISP, Nouzilly, France

☉ These authors contributed equally to this work.
* david.pasdeloup@inrae.fr

## Abstract

Inter-individual transmission is an essential part of viruses' life cycle, yet the molecular basis of transmission remains elusive. Using Marek's Disease Virus (MDV), a deadly and contagious herpesvirus of chickens that is transmitted by inhalation of contaminated dander, we previously demonstrated that tegument protein pUL47 was necessary for horizontal transmission in the natural host while being dispensable for pathogenesis and replication. We showed that pUL47 was also necessary for the correct splicing and expression of UL44 transcripts, which encode glycoprotein gC, another viral protein known as essential for MDV's transmission. Here, with the aim of characterizing the molecular basis for the role of pUL47 in transmission, we identify the cellular protein p32/C1QBP/gC1qR, a regulator of mRNA splicing, as a binding partner. We demonstrate that this interaction is necessary for transmission while being dispensable for the correct splicing and expression of UL44 transcripts, thus uncoupling the role of pUL47 in transmission from gC. These results provide a deeper molecular understanding of the natural transmission of a Herpesvirus.

## Author summary

Efficient host-to-host transmission is essential for viral success, yet knowledge regarding transmission is lacking, in part because the most relevant information regarding viral transmission is obtained *in vivo* in natural hosts. There is only a limited number of such models as costs and ethics surrounding *in vivo* experiments limit their use. Here, we use Marek's Disease Virus (MDV) and the chicken, its natural host, to shed light on the transmission of Herpesviruses, a family of viruses which is very contagious and that efficiently maintains itself in

**Data availability statement:** The full sequences of pUL47Δ71-185 as sequenced by Illumina and ONT were deposited on European Nucleotide Archive (ENA, https://www.ebi.ac.uk/ena/browser/home) and are accessible under project number PRJEB78547.

**Funding:** This work was partly funded by a grant from the Department for Animal Health (SA) of INRAE to DP. MD was supported by a PhD fellowship from the French Ministry for Higher Education and Research. A.C was supported by an INRAE/Région Centre Val-de-Loire fellowship. The funders had no role in study design, data collection and analysis, decision to publish, or preparation of the manuscript.

**Competing interests:** The authors have declared that no competing interests exist.

almost every species of the animal kingdom. We focus on viral protein pUL47, an RNA-binding protein which is abundant in the viral particle and that is necessary for transmission while being dispensable for pathogenesis. We demonstrate that pUL47 interacts with host protein p32/C1QBP/gC1qR, a known regulator of splicing, and that this interaction is necessary for natural transmission of MDV. This is a rare illustration of how a binary interaction critically affects an aspect of virus life cycle as essential as transmission without affecting pathogenesis. It also opens new and original lines of investigation on the molecular basis of viral transmission that do not include envelope proteins.

## Introduction

Host-to-host transmission is essential for viruses to be effectively maintained in the host population. Yet, the knowledge of the molecular basis for transmission is often lacking and is usually limited to the role of viral envelope proteins for cellular tropism and entry. Herpesviruses are excellent candidates to study transmission as they are very contagious and usually very restrictive in their host range, indicating that their transmission is quite selective. Transmission is best studied in the natural host of the virus. Gallid alphaherpesvirus 2 (GaAHV-2) or Marek's Disease Virus (MDV) is the prototypic virus from the *Mardivirus* genus of the *Alphaherpesvirinae* subfamily. It is an oncogenic and deadly virus of chickens which is transmitted by inhalation of contaminated dander or dust. Contrary to most alphaherpesviruses, MDV is strictly cell-associated in cell culture. *In vivo*, the virus infects CD4+T lymphocytes where its genome is integrated into telomeres of the host genome during latency [1–3]. Viral replication is most active in the feather follicle epithelium (FFE) [4] from where the virus is shed continuously during the entire life of the host. Particles associated with dander are infectious for several months in the environment. The molecular determinants for viral transmission, which includes viral release from the FFE and the efficient primary infection of immune cells in the lung [5], are limited to three viral proteins so far: pUL13 (a protein kinase encoded by the UL13 gene), glycoprotein gC (UL44) and pUL47 (UL47) [6–9]. UL44 is one of the few viral genes whose transcripts are alternatively spliced. Three UL44 transcripts were described: one unspliced, encoding full-length gC, including its C-terminal transmembrane domain (named gC-full thereafter) and two encoding two differently truncated forms of gC lacking the transmembrane domain and secreted (gC-104 and gC-145) [10]. All three forms of gC are important for viral transmission but none is strictly indispensable on its own [10]. In addition to these proteins, it was proposed that ICP27 could be involved in MDV transmission in chickens. However, viral replication was strongly impaired in the absence of ICP27, thus suggesting that transmission could be inefficient because of poor viral replication and excretion [11]. In contrast, although pUL47 is a major tegument protein, its absence had no measurable effect on viral replication in cell culture. *In vivo*, a virus lacking UL47 was as pathogenic as the parental virus and was normally shed but its horizontal transmission was completely absent, suggesting

that pUL47 could have an early function during virus entry into a naïve host that would be essential to successfully initiate infection [7]. Interestingly, we showed that in the absence of pUL47, splicing of UL44 was altered and that expression of secreted gC was impaired in infected cells. Similarly, ICP27 specifically inhibited the splicing that yields to gC-145 and it was indispensable for the expression of secreted gC [7,11]. Altogether, these observations suggest that ICP27, pUL47 and gC are interconnected and that the role in transmission of pUL47 and, to a lesser extent ICP27, could be linked to gC. In this report, in order to identify the molecular pathways mobilized by pUL47 for transmission, we looked for host proteins recruited by pUL47 and whether they are important for MDV horizontal transmission.

## Results

### MDV pUL47 interacts with p32/C1QBP/gC1qR

To identify cellular binding partners of MDV pUL47, two yeast two-hybrid screens were carried out using pUL47 as bait and two different chicken cDNA libraries from tissues of direct relevance for MDV biology: one from the Thymus (a source of T-lymphocytes which are the prime target for MDV latency) and one from feathers (a source of FFE). To our surprise, only one specific interactor was identified independently in the two separate screens, the cellular protein p32/C1QBP/gC1qR (named "p32" thereafter), a multifunctional 278 aa-long protein that was previously identified as a binding partner of pUL47 from Herpes Simplex Virus 1 (HSV-1) [12]. The sequence obtained from the thymus cDNA library (clone D30) encompassed residues 50-278 of chicken p32 (highlighted in yellow and green in Fig 1A) whereas the sequence obtained from the feather library (clone M9 and named "p32(M9)" thereafter) encompassed residues 76-278 (in yellow in Fig 1A). This indicates that the first 76 residues of p32 are dispensable for binding of pUL47. Interestingly, the first 73 residues of human p32 are absent from the mature protein after cleavage of a mitochondrial targeting sequence (MTS) (arrows in Fig 1A) [13]. Since the protein is well conserved between human and chicken (70% identity), it is plausible that a similar cleavage takes place with chicken p32. Alignment of the two sequences shows that although the cleaved domain is poorly conserved, the rest of the protein is strongly conserved (Fig 1A). Assuming that chicken p32 is cleaved at a similar position as human p32, p32(M9) is very close to the mature, cleaved form of p32 (arrows in Fig 1A). Therefore, p32(M9) is not expected to be addressed to mitochondria, as opposed to the full-length protein. Interestingly, interaction with p32(M9) is conserved with pUL47 homologues from *Gallid alphaherpesvirus 3* (GaAHV-3) strain SB1 or *Meleagrid alphaherpesvirus 1* (also named Herpesvirus of Turkey (HVT)), two other Mardiviruses which are used as vaccines against MDV (Fig 1B).

p32 was described as a subunit of splicing factor SRSF1 (also named ASF/SF2) [14], a function that was of potential interest with regards to the role of pUL47 in the splicing of UL44 and its RNA-binding properties [7,15]. The interaction between pUL47 and p32(M9) was tested by co-immunoprecipitation in transfected avian cells (Fig 1C and 1D). This showed that myc-UL47 could precipitate V5-p32 (lane 1) and that myc-p32 could reciprocally precipitate V5-pUL47 (lane 4). The interaction was specific since myc-p32 failed to efficiently precipitate two other V5-tagged viral proteins, V5-UL46 and V5-pUL32 (lanes 5 and 6) and reciprocally (lanes 2 and 3). Immunofluorescence analysis of cells expressing pUL47 alone showed that pUL47 was localized in multiple nuclear punctate domains (Fig 2A) [16]. This localization is very similar to the one reported for pUL47 from HSV-1 [15,17] or Bovine Herpesvirus-1 (BoHV-1) [18], although the nature of these structures was not identified. Interestingly, GFP-p32(M9) colocalized in these structures with pUL47 not only from MDV RB-1B (Fig 2B) but also from other Mardiviruses such as HVT and GaAHV-3 SB1 (Fig 2C). We then compared the localization of endogenous p32 to p32(M9) in the presence or absence of pUL47. Endogenous p32 was found exclusively in mitochondria in untransfected cells (Fig 3A) or in cells expressing pUL47 alone (Fig 3B). However, although the anti-p32 antibody could recognize cytoplasmic GFP-p32(M9) in transfected cells, including accumulated forms of it (Fig 3C, yellow arrows), it failed to label the nuclear accumulations of p32(M9) (Fig 3C, white arrows), as opposed to free nuclear forms, thus indicating that this antibody is unable to detect the potential endogenous nuclear accumulations of p32 where pUL47 localizes.

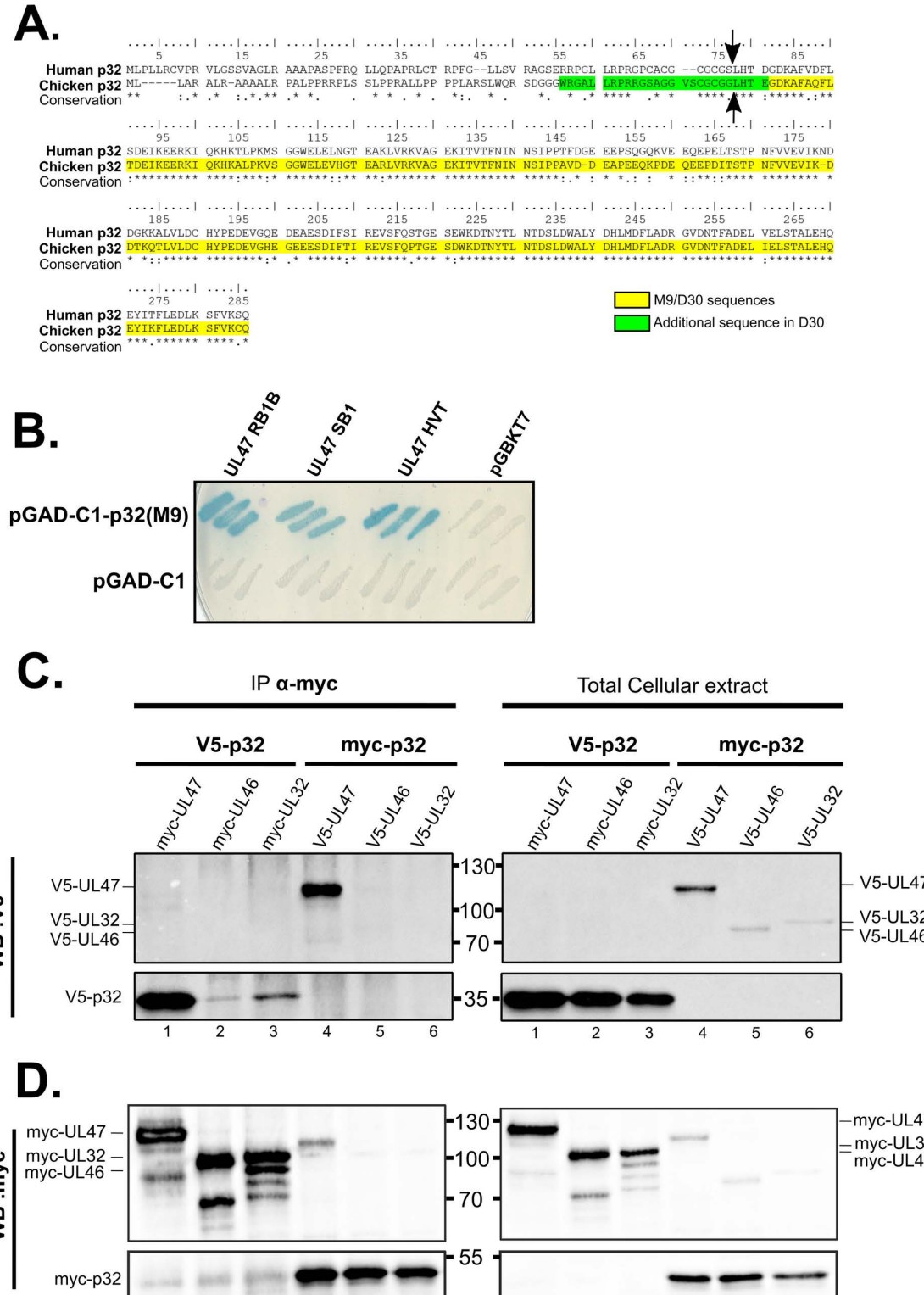

**Fig 1. pUL47 interacts with p32/C1QBP/gC1qR. (A)** Protein sequences of human and chicken p32 were aligned using CLUSTALW software. The shortest sequence isolated from Y2H (clone M9) is highlighted in yellow whereas the additional residues present in the longest sequence (clone D30)

are highlighted in green. Arrows indicate the site of cleavage of the N-terminal peptide during protein maturation (based on human p32). Residues that are completely conserved between the two sequences are shown with an asterisk (*) in the conservation line. **(B)** Y2H assay showing the interaction between p32(M9) and pUL47 from MDV (RB-1B), GaAHV3 (SB1) or HVT. Three independent colonies of yeast transformed with the indicated constructions were tested per condition. Blue coloration indicates interaction. pGBKT7 is the vector in which pUL47 was cloned and pGAD-C1 is the vector where p32(M9) is present. The two empty vectors were used as negative controls. **(C,D)** Avian cells (ESCDL-1) were transfected with plasmids expressing p32(M9) associated to a N-terminal V5 or a myc tag and plasmids expressing pUL47, pUL46 or pUL32 associated to a N-terminal myc or a V5 tag. Proteins were immunoprecipated using an anti-myc antibody and the result was analyzed by western-blot by probing membranes with an anti-V5 antibody **(C)** and subsequently re-probing the membranes with an anti-myc antibody to control for efficient immunoprecipation of myc-tagged proteins **(D)**. Western-blots corresponding to the immunoprecipitates are shown on the left whereas western-blots on total cell extracts are shown on the right. Molecular weights (in kDa) are indicated in between gels.

Taken together, these results show that pUL47 interacts with the nuclear form of p32 and that the two proteins are present in nuclear structures.

## Domain 71-185 of pUL47 is necessary for interaction with p32

We used the yeast two-hybrid system to map the minimal domain of pUL47 necessary for its interaction with p32. For this purpose, pUL47 was divided into five domains: 1-579 and 309-808 and 1-309, 309-579 and 579-808 (Fig 4A, lines 2–6). Only domains 1-579 (line 5) and 1-309 (line 2) conserved the ability to interact with p32. Therefore, domain 1-309 was further subdivided until the minimal domain of 71-185 was identified (line 12), albeit with reduced signal compared to the parental 1-185 domain (line 7). Domains shorter than 71-185, such as 71-119 or 119-185 failed to interact with pUL47 (Fig 4, lanes 10–11). Interestingly, domain 71-185 contains several serine-arginine (SR) repetitions that are reminiscent of SR repeats found in some splicing factors such as ASF/SF2, the binding partner of p32. To determine their role in binding of p32, several SR repeats were mutated to double alanine (AA) and the impact of the mutation on the binding of pUL47 to p32 was evaluated by liquid β-galactosidase assay, which is more sensitive than plate reading (Fig 4B and 4C). This assay confirmed that the β-galactosidase signal obtained with domain 71-185 alone represented less than 20% of the signal obtained with full-length pUL47 (Fig 4C), which indicates that other domains of the protein are required for optimal binding. Only the mutation of SR motifs 3 and 4 on 71-185 gave a significant reduction of the β-galactosidase signal (from 16% to less than 4%, lines 15, 16 and 20). When these repeats were mutated onto full-length pUL47, reduction of β-galactosidase signal accounted for around 30% (lines 22 and 23) showing that although some SR residues of pUL47 are important for binding of p32, they are globally dispensable. More importantly, deletion of the 71-185 domain completely abolished interaction with p32 (line 24), indicating that this domain is essential for binding to p32. Since no domain shorter than 71-185 was able to bind p32 in Y2H and that mutations of SR motifs were not sufficient to strongly impair binding of pUL47 to p32, the phenotype of pUL47Δ71-185 was further investigated.

## Deletion of domain 71-185 of pUL47 does not abrogate functions of pUL47 other than binding to p32

To verify that deletion 71-185 did not alter the proper folding of pUL47 thus invalidating the protein, a plasmid expressing pUL47 depleted of the domain and fused to a V5-tag (pV5-UL47Δ71-185) was transfected into chicken cells to verify its proper localization. The pUL47Δ71-185 protein localized to the nucleus similarly to unmodified pUL47 (Fig 5A) and, when co-expressed with GFP-p32(M9), colocalized with it in nuclear spots (Fig 5B) as was observed with full length pUL47. To confirm the Y2H results that deletion of domain 71-185 strongly impacts the interaction of pUL47 with p32, the interaction between pUL47Δ71-185 and p32 was tested in chicken cells by co-immunoprecipitation assays. This showed that pUL47Δ71-185 could precipitate only minimal amounts of p32 in co-immunoprecipitation assays as compared to unmodified pUL47 (Fig 5C), thus supporting the observations made with Y2H. Furthermore, the interaction was not mediated by RNA since addition of nuclease resulted in a comparable immunoprecipitation efficiency of p32 by pUL47 (Fig 5C). The state of RNA in cell lysates treated or not with nuclease was analyzed on an agarose gel to verify that the nuclease treatment was efficient, using purified RNA as a positive control (Fig 5D). This showed that most of RNA was degraded during cell lysis, including in the absence of nuclease, thus further ruling out a possible role of RNA in mediating the interaction.

**A.**

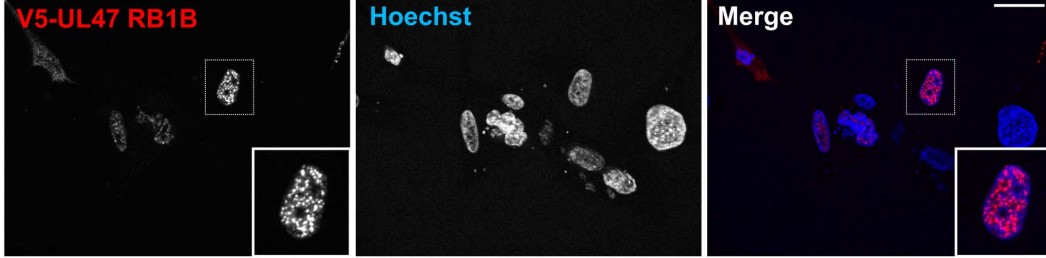

**B.**

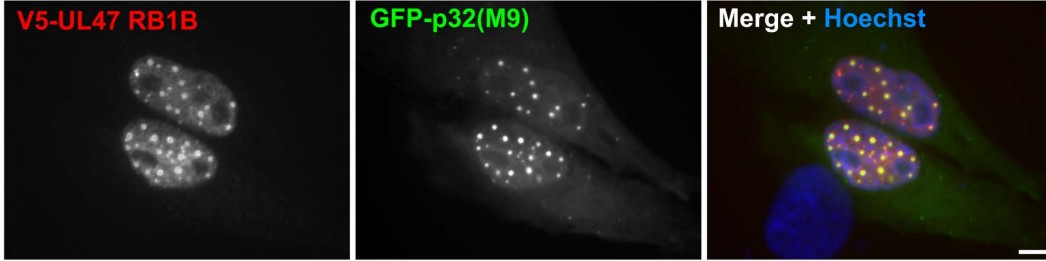

**C.**

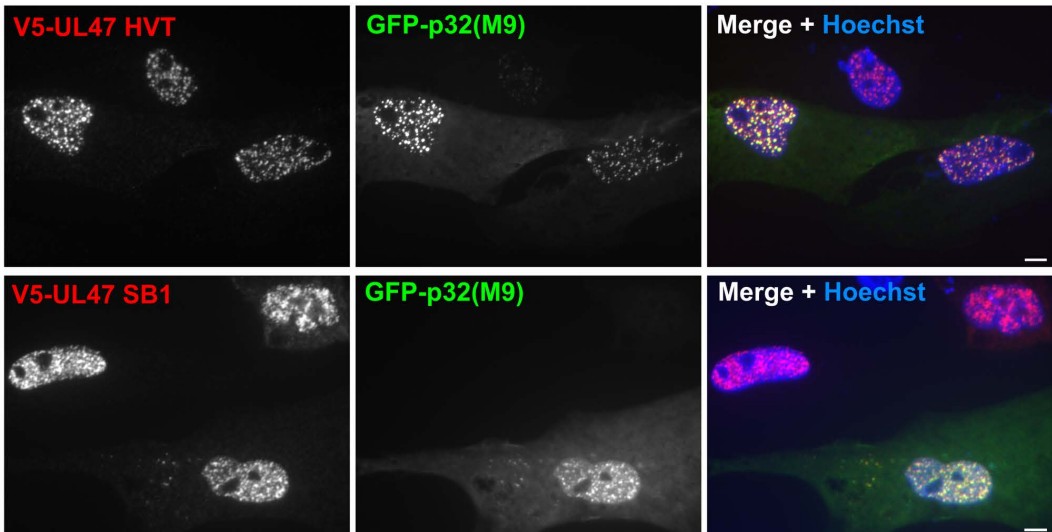

**Fig 2. pUL47 and p32 co-localize in nuclear structures.** ESCDL-1 cells were transfected with a plasmid encoding V5-pUL47 from RB-1B **(A,B)** or from HVT or SB1 **(C)** with a plasmid expressing p32(M9) N-terminally fused to GFP **(B, C)** or alone **(A)**. pUL47 was detected using an anti-V5 primary antibody and a goat anti-mouse AlexaFluor 594-conjugated secondary antibody (red). Scale bars: 5 µm.

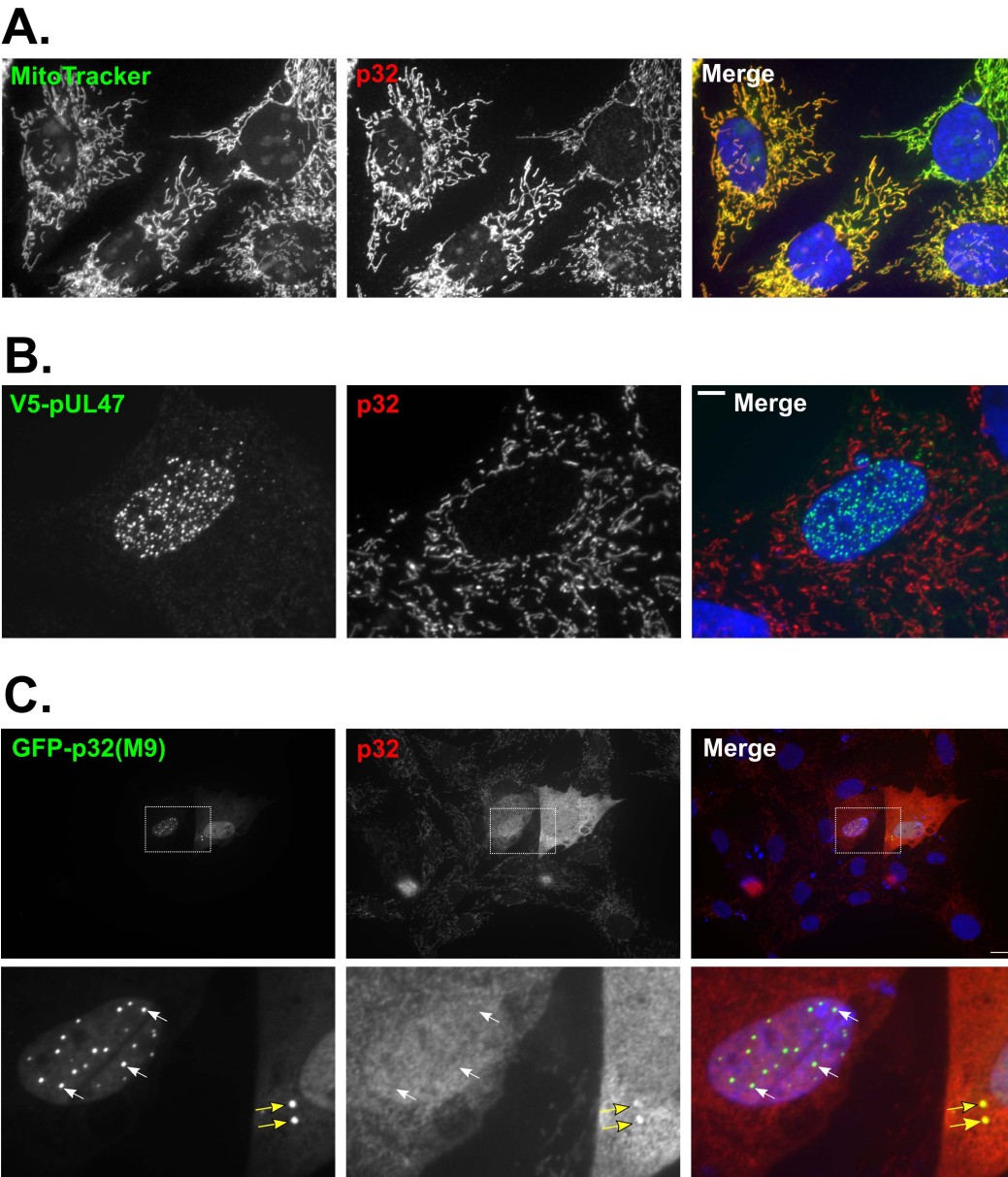

**Fig 3. Nuclear p32(M9) structures are not detected by an anti-p32 polyclonal antibody whereas cytoplasmic p32(M9) is.** Detection of endogenous p32 by an anti-p32 polyclonal antibody in non-transfected ESCDL-1 cells counterstained with Mitotracker to label mitochondria **(A)** or in cells transfected with a plasmid encoding V5-pUL47 **(B)** or GFP-p32(M9) **(C)**. **(A)** A goat anti-rabbit AlexaFluor 488-conjugated antibody was used for secondary detection of p32 (pseudo-colored in red) and mitochondria were labelled with Mitrotracker Orange (pseudo-colored in green). **(B-C)** A goat anti-rabbit AlexaFluor 594-conjugated antibody was used for secondary detection (red) of endogenous p32 whereas pUL47 was detected using an anti-V5 primary antibody and a goat anti-mouse AlexaFluor 488-conjugated secondary antibody. Note that endogenous mitochondrial p32 is hardly visible in pictures shown in **(C)** because of the stronger signal from cells overexpressing GFP-p32(M9). White arrows indicate examples of GFP-positive nuclear structures that are not detected by the anti-p32 antibody, whereas yellow arrows indicate similar GFP-positive cytoplasmic structures that are detected by the antibody. Scale bars: 20 µm for global picture in C, 5 µm for others.

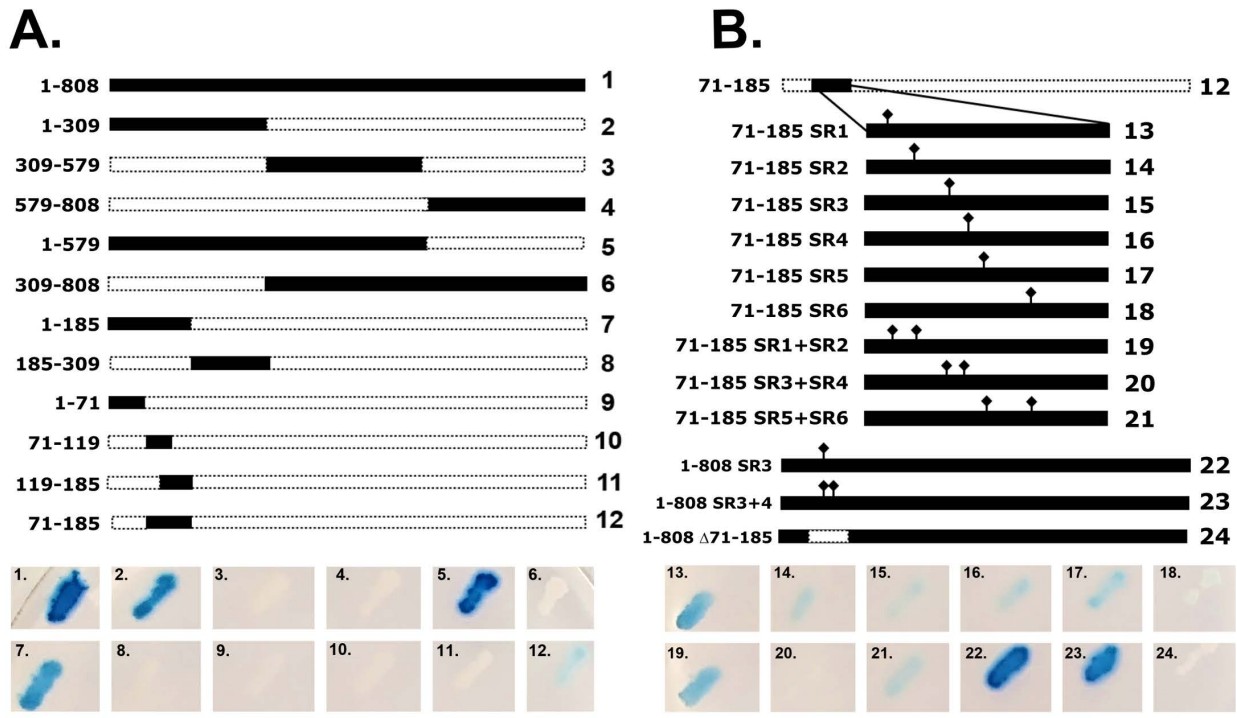

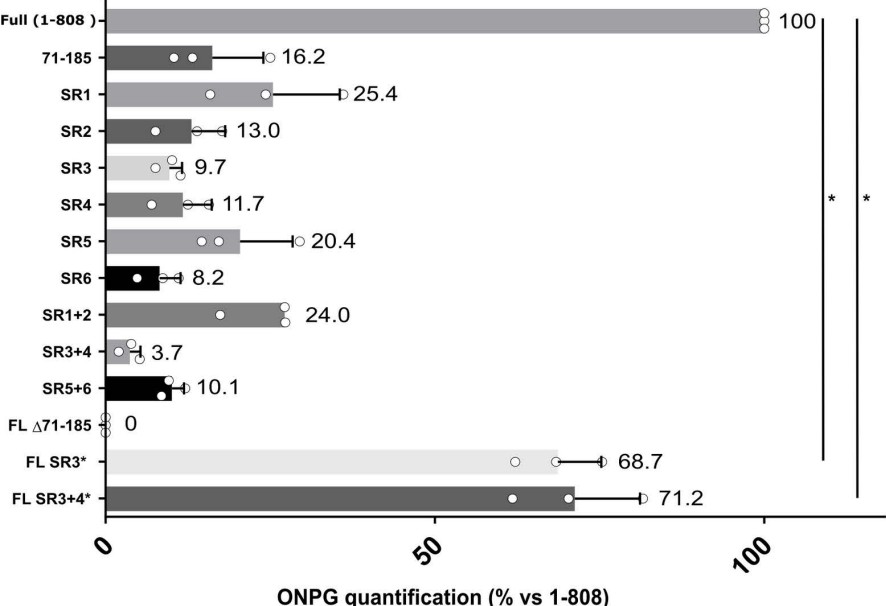

**Fig 4. pUL47 interacts with p32 via the 71-185 domain of pUL47. (A)** Schematic listing of domains of pUL47 tested in Y2H for interaction with p32 (top) with corresponding β-galactosidase coloration (bottom). The full-length protein is 808-aa long. **(B)** Schematic listing of single mutations within the 71-185 domain of pUL47 tested in Y2H with p32 and of some of these mutations in the full-length protein (top) with corresponding β-galactosidase coloration (bottom). **(C)** Quantification of liquid β-galactosidase coloration in yeasts induced by the interaction of a subset of mutants with p32, relatively to the full-length protein (1-808). Measurements were done in triplicates over three independent experiments. *:p<0.5 using a Wilcoxon-Mann-Whitney test.

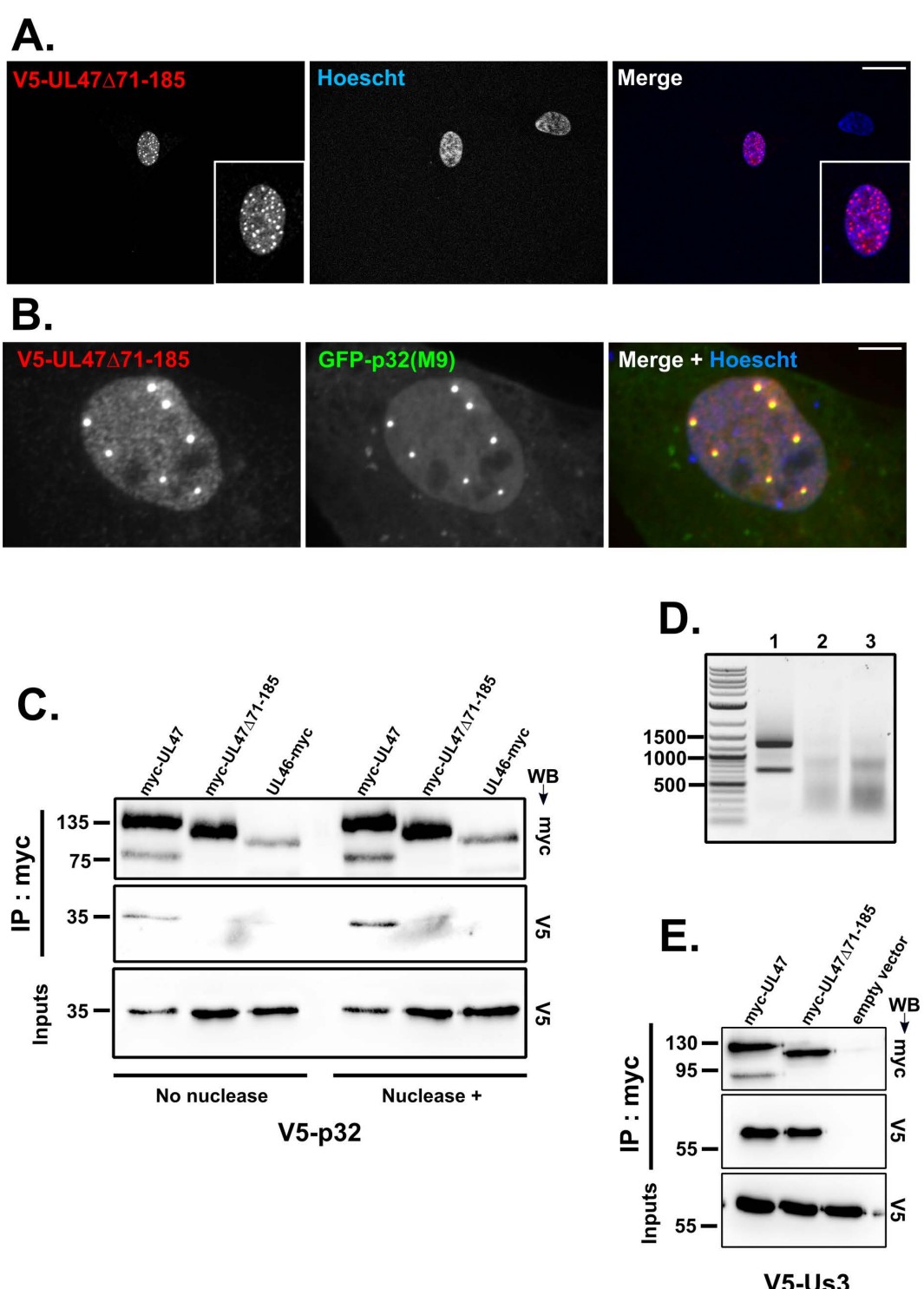

**Fig 5. pUL47Δ71-185 interacts poorly with p32 in chicken cells but normally with pUs3 and colocalizes with p32 in the nucleus. (A,B)** ESCDL-1 cells were transfected with a plasmid encoding V5-pUL47Δ71-185 without **(A)** or with a plasmid expressing p32(M9) N-terminally fused to GFP **(B)**. pUL47 was detected using an anti-V5 primary antibody and a goat anti-mouse AlexaFluor 594-conjugated secondary antibody (red). Scale bars: 5 µm. **(C,E)** ESCDL-1 cells were transfected with plasmids expressing p32(M9) **(C)** or pUs3 **(E)** associated to a V5 tag and plasmids expressing pUL47, pUL47Δ71-185 or pUL46 associated to a myc tag or an empty pcDNA3-myc vector. Cells were lysed 24h post-transfection and micrococcus nuclease

was added or not in the lysate. Proteins were immunoprecipitated using an anti-myc antibody and the results were analyzed by western-blot using an anti-V5 and an anti-myc antibody. Molecular weights (in kDa) are indicated on the left side of each gel. **(D)** 500 ng of purified RNA (lane 1) or RNA purified from the lysates of cells expressing myc-pUL47 and V5-p32(M9) untreated (lane 2) or treated (lane 3) with micrococcal nuclease were analyzed on a 0.8% agarose TAE gel. Molecular weights (in bp) are indicated on the left side of the gel.

In order to test whether the loss of interaction between pUL47Δ71-185 and p32 could be linked to a non-functional pUL47 protein, viral kinase pUs3, a known interactor of pUL47 for HSV-1 [19], was used as a control. In contrast to p32, pUL47Δ71-185 could immunoprecipitate pUs3 as efficiently as unmodified pUL47 (Fig 5E). Altogether, these results demonstrate that deletion of domain 71-185 of pUL47 is sufficient to abolish the interaction with p32 without invalidating pUL47 localization and its binding to pUs3, thus suggesting that this deletion does not affect pUL47 functions other than binding to p32.

### Genomic and phenotypic characterization of vUL47Δ71-185

In order to test the role of p32 during the viral cycle of MDV, deletion of domain 71-185 of UL47 was introduced in the genome of very virulent strain RB-1B of MDV, thus generating recombinant genome rUL47Δ71-185. Genome integrity of rUL47Δ71-185 was verified by restriction fragment length polymorphism (RFLP), which showed to have a comparable profile to the one of parental rRB-1B while confirming the presence of the deletion (Fig 6A). rUL47Δ71-185 was fully sequenced by two different technologies of next-generation sequencing (NGS) (Illumina and Oxford Nanopore (ONT)) to check for possible additional mutations that could have been introduced during the mutagenesis process (Fig 6B). Mutations that were identified by Illumina sequencing and confirmed by ONT are listed in Fig 6B. They include deletion 71-185 in UL47 as the only mutation affecting a viral gene. Upon transfection of primary chicken embryonic skin cells (CESCs) with rUL47Δ71-185, infectious virus was recovered and named vUL47Δ71-185 thereafter. Spread of vUL47Δ71-185 was tested in primary embryonic skin cell culture by measuring viral plaque size and was found to be comparable to parental vRB-1B (Fig 6C), as was observed previously with vΔUL47 [7].

### pUL47Δ71-185 does not localize in nuclear speckles in infected cells and has delayed nuclear import kinetics

To analyze the properties of pUL47Δ71-185 in infected cells, we derived vUL47Δ71-185-GFP from vUL47Δ71-185 where GFP was fused to the C-terminus of pUL47. vUL47-GFP was similarly derived from parental vRB-1B. CESCs were infected with either virus and pUL47-GFP localization was monitored and compared to that of capsid protein VP5, a marker of viral replication centers in the nucleus, and endogenous p32 (Fig 7). Surprisingly, while pUL47-GFP localized in nuclear punctates as was observed in transfected cells (Fig 7A), pUL47Δ71-185-GFP's localization was nucleo-cytoplasmic and no nuclear punctates could be observed (Fig 7B). Of note, nuclear accumulations of pUL47-GFP were generally distinct from capsid replication compartments and infected cells displayed poor p32 staining, as opposed to neighboring non-infected cells. Treatment of cells with leptomycin B (LMB), a drug that inhibits CRM1-dependent nuclear export [20,21], induced accumulation of pUL47 in the nucleus, independently of the mutation. This demonstrates that pUL47 is capable of nuclear export in infected cells and that deletion of domain 71-185 does not affect this property.

To further examine MDV pUL47's nucleocytoplasmic shuttling properties, FLIP (Fluorescence Loss In Photobleaching) experiments were carried out in living infected cells. In these experiments, pUL47GFP was locally and repeatedly photobleached in the cytoplasm, thus gradually depleting the pool of fluorescent cytoplasmic protein over a period of ten minutes (Fig 8A). Total nuclear fluorescence was then quantified and normalized to fluorescence at T0 in bleached (FLIP+) cells and in neighboring unbleached cells (FLIP-) as control for global bleaching linked to repeated scanning (Fig 8B). The time required for the loss of 50% of nuclear fluorescence ($t_{1/2}$) was then determined (Fig 8C) as an indicator of nuclear import dynamics. We observed that nuclear fluorescence declined in cells infected with either virus, indicating

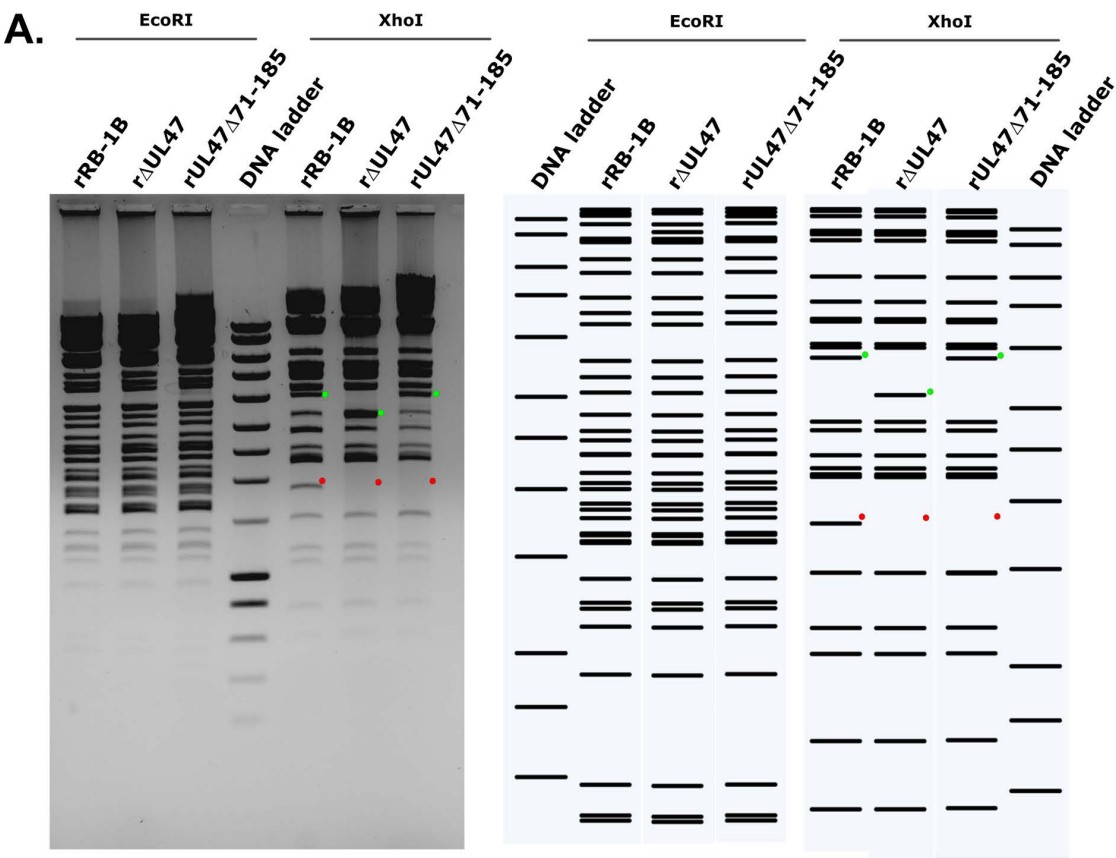

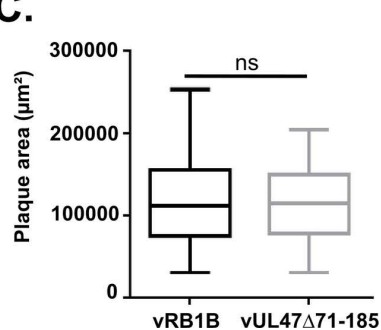

**Fig 6. Characterization of vUL47Δ71-185 in vitro. (A)** RFLP analysis of rUL47Δ71-185 in comparison to parental rRB-1B and rΔUL47. 1 µg of purified BACs were digested with EcoRI or XhoI. The digestion profiles were analyzed on a 0.6% TBE agarose gel (left) and compared to the corresponding expected profiles (right). The DNA ladder used is SmartLadder (Eurogentec). Red and green dots indicate specific bands. **(B)** Summary of NGS

| RB1B | Illumina U$_L$47Δ71-185 | ONT check | Gene involved | Sanger analysis |
|---|---|---|---|---|
| 4441 | Deleted C in homopolymeric region | Confirmed | None (see 137890) | ND |
| 109732 | Deletion of 345 nt (Δ71-185 in UL47) | Confirmed | UL47 | Confirmed |
| 137890 | Deleted G in homopolymeric region | Confirmed | None (see 4441) | ND |
| 158243 | Insertion BAC sequence (7.2 kb) | Confirmed | None | ND |

analyses of rUL47Δ71-185 including the results of Illumina and ONT sequencing and corresponding Sanger sequencing. Reference genome used was *Gallid herpesvirus 2* strain RB-1B complete genome (accession number EF523390). ND = not done. **(C)** Plaque size assay of vUL47Δ71-185 in CESCs as compared to parental vRB-1B. Significant differences in the median of plaque areas were determined using a Wilcoxon-Mann-Whitney test. 50 plaques per condition were measured on a single experiment. ns: not significant.

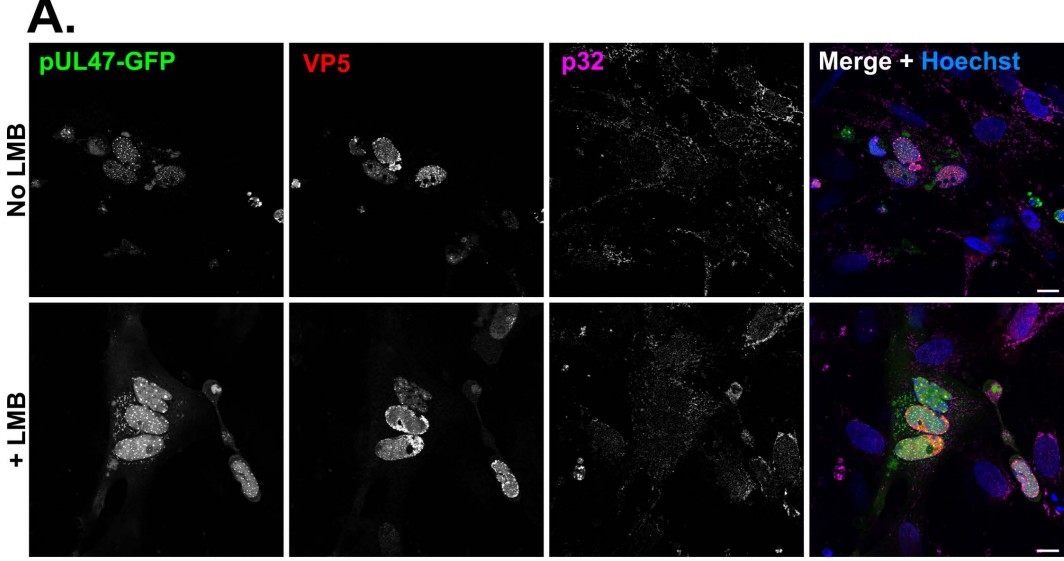

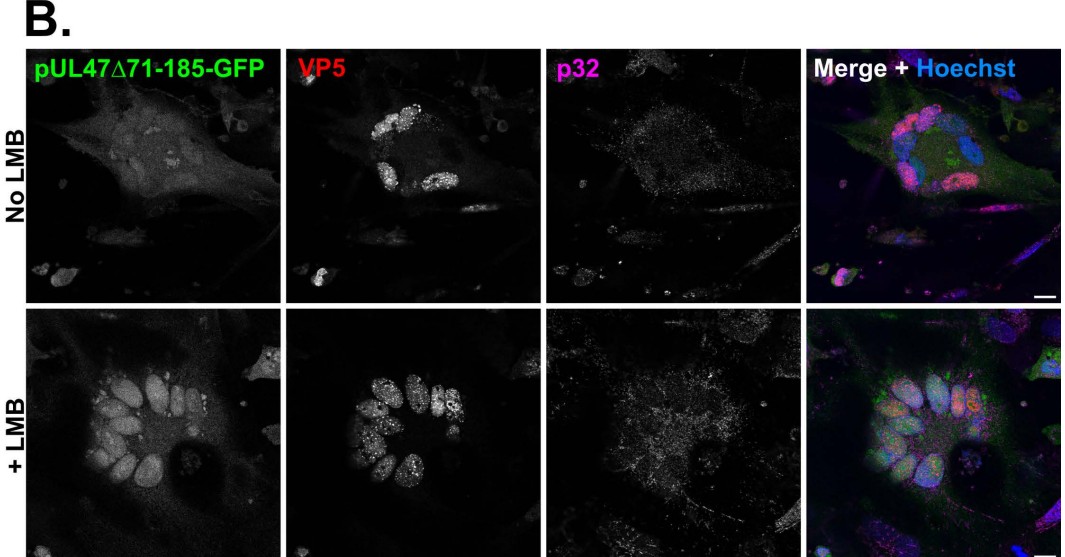

**Fig 7. pUL47Δ71-185 is capable of transport from the nucleus to the cytoplasm but does not localize in nuclear punctates in infected cells.** Chicken Embryonic Skin Cells (CESCs) were infected with vUL47-GFP **(A)** or vUL47Δ71-185-GFP **(B)** in the presence or absence of 50nM of Leptomycin B (LMB) and were labelled with VP5-specific monoclonal antibody F19 and a goat anti-mouse AlexaFluor 555-conjugated secondary antibody (red) and a p32-specific antibody and a goat anti-rabbit AlexaFluor 633-conjugated secondary antibody (magenta). pUL47 was visualized through direct GFP fluorescence (green). Scale bars: 10 μm.

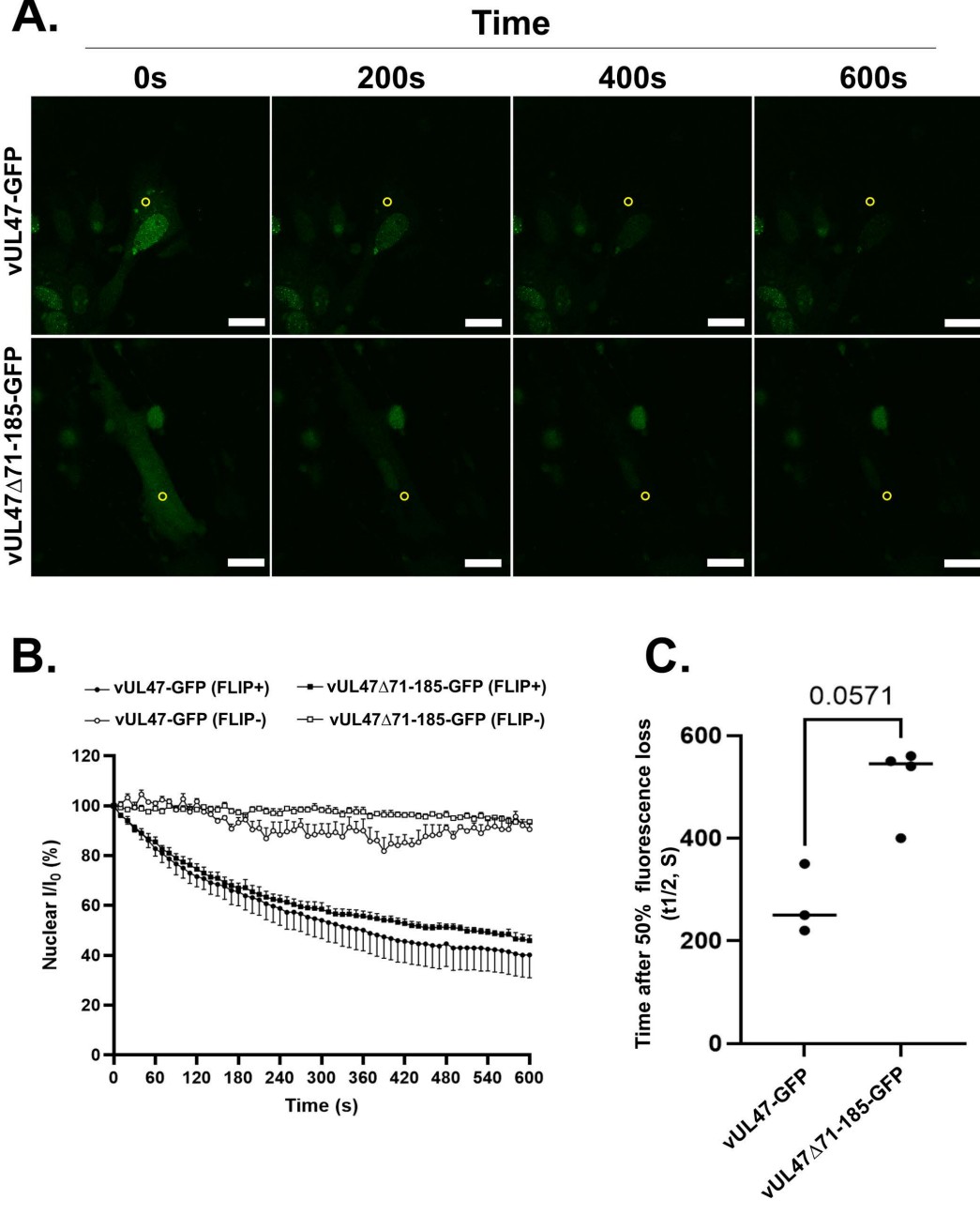

**Fig 8. pUL47Δ71-185 nuclear import is impaired in infected cells.** Chicken Embryonic Skin Cells (CESCs) were infected with vUL47-GFP or vUL47Δ71-185-GFP and GFP fluorescence was monitored live four days-post-infection. **(A)** A 4μm diameter circle (yellow circle) in the cytoplasm was repeatedly photobleached with a 488nm laser every 10 seconds for a total duration of 10 minutes and cells were imaged before every bleaching event. Scale bars: 20μm. Nuclear fluorescence intensity (I) of bleached-(FLIP+) and unbleached-cells (FLIP-) in the same image was then quantified over time and normalized to the nuclear intensity at T0 (I0)**(B)** so that time required for loss of 50% of nuclear fluorescence (t1/2) could be estimated **(C)**. A Wilcoxon-Mann-Whitney test was used to test for significance.

that both pUL47-GFP and pUL47Δ71-185-GFP are capable of nuclear import. However, decline of pUL47Δ71-185-GFP nuclear fluorescence ($t_{1/2} = 550s$) was markedly slower than that of pUL47-GFP ($t_{1/2} = 250s$), indicating that nuclear import of pUL47Δ71-185-GFP is less dynamic than pUL47-GFP. Altogether, these results demonstrate that, contrary to pUL47-GFP, pUL47Δ71-185-GFP cannot localize to nuclear punctates in infected cells and that although the mutated protein is capable of nucleocytoplasmic shuttling, its nuclear import is delayed compared to WT pUL47.

## Domain 71-185 of pUL47 is necessary for horizontal transmission of MDV

Since the only phenotype that could be associated with vΔUL47 was a total absence of horizontal transmission between animals [7], the transmission of vUL47Δ71-185 was tested *in vivo* following a similar protocol. 3000 PFU (plaque-forming units) of vUL47Δ71-185 or the parental vRB-1B virus were inoculated by the intramuscular route to nine White Leghorn chickens of two-weeks of age. Eight naïve contact chickens were added to each group to test for natural transmission of the virus. As shown in Fig 9A, Marek's Disease (MD) incidence and kinetics of disease were similar between animals inoculated with vRB-1B or vUL47Δ71-185 with 100% MD incidence reached at 46 and 43 days post-inoculation respectively and a median of survival of 32 days for both groups. At necropsy, all inoculated animals had tumors (Fig 9B). Interestingly, no contact animal from the vUL47Δ71-185 group developed the disease by the end of experiment. In contrast, 50% of contact animals in the vRB-1B group displayed clinical signs of MD and were consequently euthanized before the end of the experiment. All these animals had tumors, as well as one out of the five symptom-free animals, as opposed to contact animals from the vUL47Δ71-185 group where no tumor was found at necropsy at the end of the experiment (Fig 9B). Viral genome loads were measured in peripheral blood mononuclear cells (PBMC) and feathers by quantitative PCR (qPCR) of inoculated animals. They were found in statistically comparable amounts between the two groups at 14 and 21 days post-infection (dpi) both in PBMCs and feathers (Fig 9C). In contrast, while viral loads in PBMC of contact chickens of the vRB-1B group were in the same order of magnitude as inoculated animals, viral loads of the vUL47Δ71-185 group were under or close to the background threshold up to the end of the experiment, thus indicating a failure in viral transmission (Fig 9D). To test whether this lack of transmission was associated with poor viral excretion from inoculated animals, viral loads were quantified from dust where contaminated material is expected to accumulate. Accumulated dust was collected weekly in each isolation unit (between days 21–28, 28–35 and 35–42 pi) and viral loads were quantified (Fig 9E). This showed that viral loads in dust were comparable between the two groups although a decrease was observed in the vUL47Δ71-185 group between days 35 and 42 post-inoculation. This was attributed to the fact that after 28dpi, there were less infected animals shedding the virus in the vUL47Δ71-185 group since contact animals were not infected in this group and therefore did not shed virus, contrary to contact animals of the vRB-1B group. Altogether, these results show that domain 71-185 of pUL47 is necessary for the role of pUL47 in horizontal transmission of MDV.

## Domain 71-185 of pUL47 is not involved in the splicing of UL44 *in vivo*

We previously showed that in cells infected with a mutant deleted for UL47, unspliced forms of UL44 accumulated and spliced forms decreased in comparison to vRB-1B-infected cells [7]. Since all three forms of UL44 are necessary for an optimal horizontal transmission [10], we hypothesized that at least part of the function of UL47 in transmission was related to UL44 splicing, possibly via the interaction of pUL47 with p32.

To verify this hypothesis, RNA was extracted from skin and feather samples from animals infected with vRB-1B or vUL47Δ71-185 and the expression and splicing of UL44 were monitored by RT-PCR. Transcripts of UL18 were used as infection control. As shown in Fig 10, all three forms of UL44 transcripts were found in most of samples. However, in five samples (skin and feather from animal #52, skin from #66 and #67 and feather from #70), only the unspliced form of UL44 could be detected, a phenotype that was never observed in cell culture. Interestingly, it was always associated with a particularly strong expression of UL47. However, this phenotype was observed in both vRB-1B (Fig 10A) and vUL47Δ71-185 groups (Fig 10B), indicating that it is independent of the 71-185 domain of pUL47. Since there is no difference of splicing

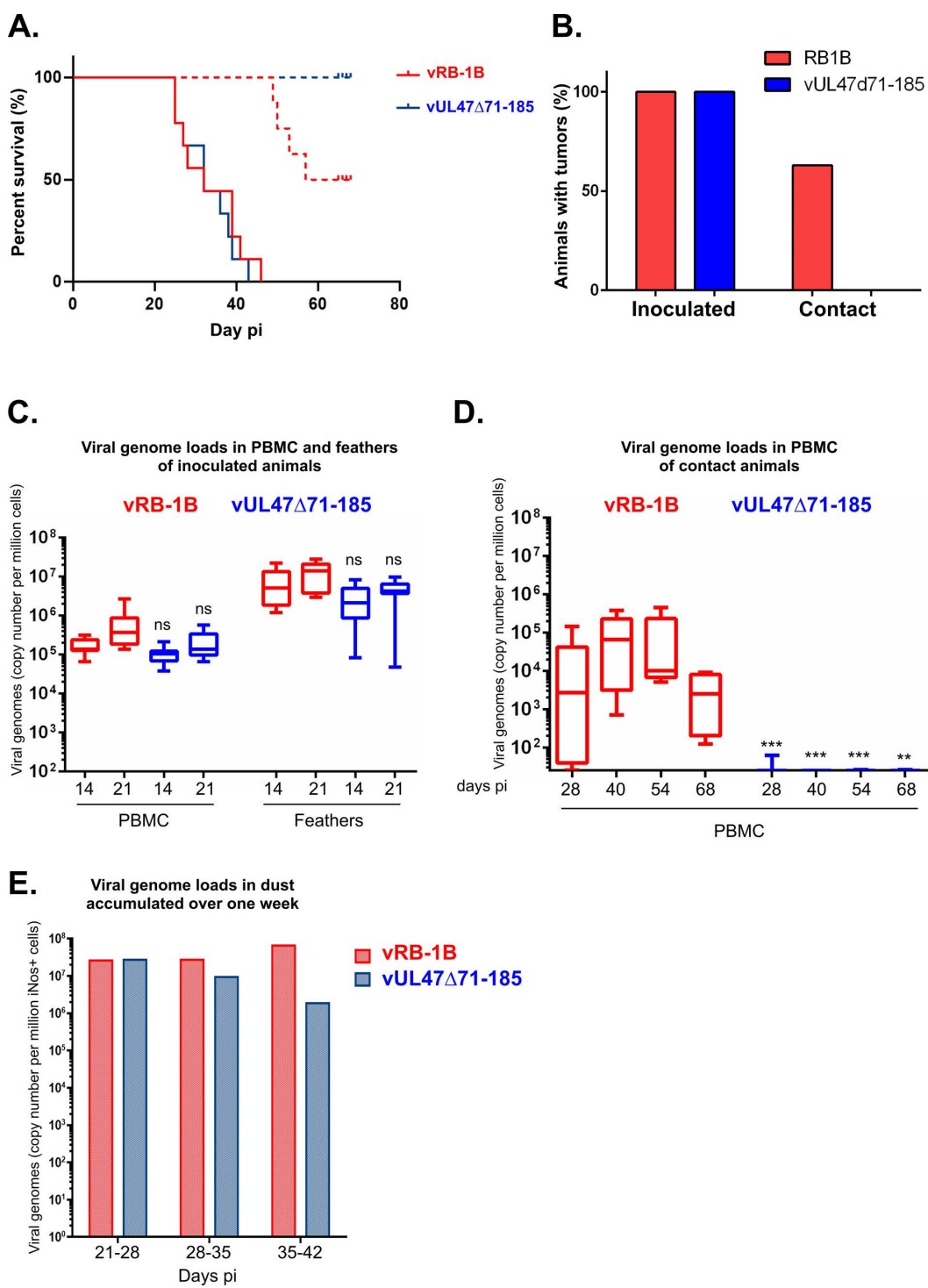

**Fig 9. vUL47Δ71-185 is unaffected in pathogenesis and is not horizontally transmitted. (A)** Nine 2-weeks old white Leghorn chickens were inoculated with 3000 pfu of vRB-1B or vUL47Δ71-185. The percentage of live animals is shown for vRB-1B (plain red line) or contact animals (dashed red line) and vUL47Δ71-185-inoculated (plain blue line) or contact animals (dashed blue line). Animals with apparent symptoms of Marek's disease were euthanized to limit suffering. **(B)** Percentage of animals with tumors visible during necropsy. None of vUL47Δ71-185 contact animals presented tumors. **(C,D)** DNA was extracted from PBMC or feathers of chickens inoculated with vRB-1B (in red) or vUL47Δ71-185 (in blue) at 14 and 21 days post-inoculation **(C)** or from PBMCs from contact chickens at 28, 40, 54 and 68 days post-inoculation **(D)**. Viral genomes were quantified by real-time

quantitative PCR and their numbers are indicated as per million of cells. Results are shown as medians with interquartile range (box) and minimal and maximal values (whiskers). Significance of differences of viral loads between vRB-1B-infected birds and vUL47Δ71-185-infected birds at each time point were determined using a Wilcoxon-Mann-Whitney test. ns: not significant (p > 0.5); **: p < 0.01; ***: p < 0.001. **(E)** The amount of viral genomes from DNA extracted from the dust accumulated during one week (between days 21 and 28 pi, between days 28 and 35 pi and between days 35 and 42 pi and collected from the two separate isolators (vRB-1B in red and vUL47Δ71-185 in blue) was determined by qPCR. Results are indicated as per million of iNos-positive cells as a reference because part of cells in dust (essentially corneocytes) are devoid of nucleus and cannot be accounted for by qPCR.

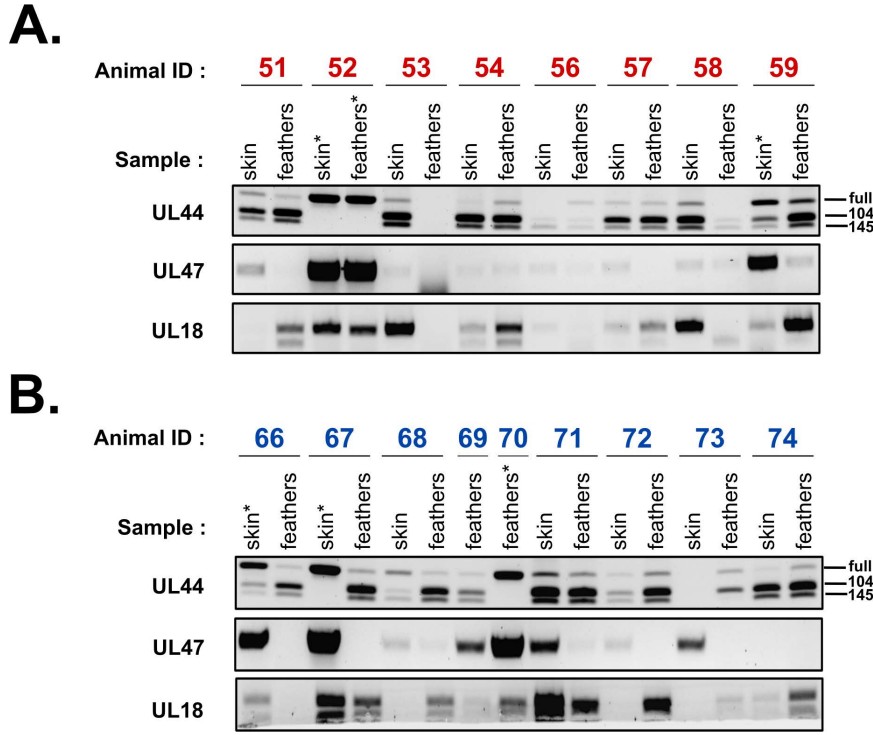

**Fig 10. The role of pUL47 in inter-individual transmission is not associated with alternative splicing of UL44.** RNA was extracted from skin or feathers samples of chickens inoculated with vRB-1B (A) or vUL47Δ71-185 (B). The profile of UL44 splicing was determined by RT-PCR using primers encompassing the splicing zone. The upper band corresponds to unspliced UL44 whereas the middle and bottom bands correspond to UL44-104 and UL44-145 respectively. Primers to detect UL47 and UL18 were also used. UL18 is used as an infection marker. (*) indicates samples where UL44 was essentially or exclusively unspliced.

of UL44 that is specific to the vUL47Δ71-185 group, we conclude that domain 71-185 of pUL47 is not involved in the splicing of UL44. Therefore, the role of domain 71-185 of pUL47 in horizontal transmission is independent of the splicing of UL44.

## Discussion

We previously made the unexpected observation that pUL47, a protein described as being abundant in the outer tegument of alphaherpesviruses [22–25], was completely dispensable for replication and pathogenesis of MDV and that the only phenotype that could be firmly associated to the absence of pUL47 was the complete lack of inter-individual transmission by the natural route, despite unaffected viral shedding [7]. A likely explanation for this phenotype is that pUL47 performs one or several early functions during entry in the host that are essential to initiate infection. This kind of phenotype is more likely to be associated with envelope proteins, as they are primarily involved in the first steps of viral invasion. Glycoprotein gC was thus demonstrated to be important for horizontal transmission and, since we showed that pUL47

was important for correct splicing of the UL44 gene coding of gC [7], we hypothesized that the absence of pUL47 affected UL44 splicing, thus undermining transmission. Here we show that the function of pUL47 in transmission requires domain 71-185 of the protein, a domain that is necessary for interaction with cellular protein p32/C1QBP/gC1qR, a multifunctional protein that was reported as a regulator of the ASF/SF2 splicing factor [14]. Although it is very likely that the interaction of p32 with pUL47 is necessary for transmission, we cannot exclude that the 71-185 domain of pUL47 could be involved in the interaction with other, yet unknown, cellular or viral proteins.

The interaction between pUL47 and p32 is conserved amongst other members of the *Mardivirus* genus (Fig 1B), including their colocalization in nuclear punctate domains (Fig 2C). It was also previously reported for HSV-1 by tandem affinity purification. In that study, the authors described the localization of p32 and pUL47 at the nuclear envelope of infected cells, presumably during viral nuclear egress [12]. Using a p32-specific antibody, we did not observe any localization of endogenous p32 other than its known association to mitochondria including in cells expressing pUL47 (Fig 3A and 3B). In addition, a role of MDV pUL47 in nuclear egress is unlikely because the impairment of nuclear egress would result in altered viral growth in the absence of pUL47 *in vitro* and *in vivo* and MDV vΔUL47 grows normally, contrary to HSV-1 or Pseudorabies Virus (PrV) [7,19,26,27].

Although binding of pUL47 to p32 remains to be shown in infected cells, we demonstrate that deletion of domain 71-185 of pUL47 was sufficient for an almost complete loss of interaction between pUL47 and p32 in yeast two-hybrid and co-immunoprecipitation assays. However, this deletion affected neither the intrinsic localization of pUL47 to nuclear spots in transfected cells (Fig 5A and 5B) nor its interaction with pUs3 (Fig 5E), an interaction previously demonstrated for HSV-1 [19] and showed here for MDV for the first time. In contrast, in infected cells, deletion of the 71-185 domain was associated to a complete loss of accumulation to nuclear spots (Fig 7) and impaired nuclear import (Fig 8), resulting in a mutated protein localizing throughout the cell as opposed to WT pUL47 which localized essentially in the nucleus and in nuclear spots. Although we were unable to identify the nature of these nuclear spots, there are reminiscent of bodies such as Cajal bodies, nuclear speckles or paraspeckles, all structures linked to gene expression, RNA maturation, export and/or splicing [28–30]. p32 is part of the ASF/SF2 splicing complex which localizes in nuclear speckles [31] and pUL47 colocalizes with nuclear p32 in non-infected cells (Fig 2) thus suggesting that pUL47 localizes to nuclear speckles. Deletion of domain 71-185 results in unmodified localization in transfected cells whereas localization to speckles is lost in infected cells, indicating that binding to p32 is dispensable for the intrinsic property of pUL47 to associate to speckles but that, during infection, binding to p32 is necessary to maintain pUL47 in speckles or effectively transport it there. Therefore, it is likely that the function of pUL47 in transmission is linked to splicing regulation in nuclear speckles and that binding to p32 is essential for this function.

pUL47 from HSV-1 was documented as an RNA-binding protein [15] and we previously reported that pUL47 can affect splicing of UL44 [7], which naturally led us to focus on p32 as a splicing regulator. RT-PCR analysis on RNA extracted from skin and feather samples of infected animals showed that the splicing of UL44 was highly variable between samples. Interestingly, we observed the absence of splicing of UL44 in a few samples, which resulted in the exclusive presence of the unspliced transcript, a phenotype that was never observed in cell culture. In these samples, UL47 transcripts were particularly abundant, suggesting a link between strong UL47 expression and inhibition of UL44 splicing. Jarosinski and collaborators described how pUL47 was poorly expressed *in vitro* but strongly expressed in the FFE, thus hinting at a possible link between pUL47 strong expression and efficient viral assembly in the FFE [32]. pUL47 could therefore have a role in the FFE by regulating expression of certain viral proteins that are key for assembly and egress differently *in vivo* and *in vitro*. The inhibition of UL44 splicing could be such a regulation, whereby only full length gC is expressed in the FFE and not the secreted forms, which could be important for viral assembly and/or shedding. However, inhibition of UL44 splicing was present in both vRB-1B and vUL47Δ71-185 groups, indicating that residues 71-185 of pUL47 are dispensable for the potential role of pUL47 on UL44 splicing. This latter observation associated to the fact that residues 71-185 are necessary for transmission show that the role of pUL47 in horizontal transmission is not associated with UL44 splicing.

Since p32 is an inhibitor of ASF/SF2 and that ASF/SF2 is involved in alternative splicing [33], it is possible that pUL47 can impact alternative splicing of cellular genes in infected cells early in infection, which could result in successful infection.

It is worth considering that, contrary to infected cells *in vitro*, pUL47 localizes in the cytoplasm of keratinocytes in the FFE of infected animals [32]. This indicates that the function of pUL47 in the skin could differ from that observed *in vitro*. For instance, there is active viral assembly in the FFE as opposed to infected cells *in vitro*, and pUL47 could perform *in vivo* its role in assembly which takes place in the cytoplasm. However, it is unlikely to be the unique role of pUL47 in the cytoplasm since the absence of pUL47 does not seem to affect viral shedding [7]. An additional possibility is that pUL47 could interact with mitochondria-associated p32 to potentiate its inhibition of MDA-5-dependent antiviral activities, as was observed with Sendai Virus [34] or using its shuttling properties [15,17] to delocalize nuclear p32 to the cytoplasm.

An explanation to the multifunctional role of p32 is that the protein can act as a chaperone involved in stabilizing other proteins with varied functions. Such a stabilizing role of p32 was reported earlier, including for viral proteins [35–37]. Given that pUL47 is a poorly expressed nuclear protein in cell culture as opposed to the FFE where it is cytoplasmic and strongly expressed, p32 could act as a cytosolic chaperone *in vivo* and thus allow accumulation of pUL47 in the FFE to promote transmission.

The primary role of p32/C1QBP/gC1qR as a receptor to complement protein C1q [13] could be of importance for controlling MDV during the early steps of infection *in vivo*. A similar interaction between Hepatitis C Virus (HCV) Core protein and gC1qR was shown to down-regulate T-cell response to infection [38–41]. T-cell mediated elimination of the first MDV-infected cells triggered by the first-line host defense mechanism, i.e., the complement system, could result in abortive infection during natural infection. pUL47 binding to p32/gC1qR could therefore result in survival of infected cells by inhibiting complement-mediated antiviral response. Interestingly, one of the first functions attributed to glycoprotein gC, which is also involved in horizontal transmission of MDV, was of an immunomodulatory protein interfering with the complement system [42,43].

Although it remains to be determined what are the exact pathways mobilized by the pUL47/p32 complex, this study shows that the essential role of pUL47 in transmission is carried by domain 71-185 which is necessary for binding p32/C1QBP/gC1qR and localization of pUL47 to nuclear speckles. The importance of these results is highlighted by the fact that the pUL47/p32 interaction is conserved not only among Mardiviruses but also for HSV-1 [12], it is therefore likely that it is essential for the transmission of all alphaherpesviruses.

## Materials and methods

### Ethics statement

*In vivo* experiments were carried out according to the guidance and regulation of the French Ministry of Higher Education and Research (MESR) and the experimental protocol was approved by the regional ethics committee, CREEA VdL ("Comité d'Éthique pour l'Expérimentation Animale Val de Loire") under APAFIS number #31071-2021041615377018.

### Yeast two hybrid: Preparation of cDNA and yeast libraries

Two yeast DNA libraries were made using a method adapted from the "Mate & Plate" system (Clontech). In this system, cDNA is recombined directly in yeast into a linearized destination vector (pGAD-C1-rec here) using recombination sequences (AAGCAGTGGTATCAACGCAGAGTGGCCATTATGGCC) present in both the vector and the cDNAs. pGAD-C1-rec confers prototrophy to Leucine to yeast and fuses the GAL4-activation domain (AD) at the N-terminus of the protein encoded by the cloned cDNA sequence.

Total RNA was extracted from a sample of thymus or feather follicle material (feather wall and pulp) from a 53-days old White Leghorn hen using Trizol and mRNA was subsequently purified using the NucleoTrap mRNA kit (Macherey-Nagel). mRNA was used as a template for first-strand reverse transcription using MMLV reverse transcriptase (Promega) and the following primers: polyT_CDSPrimer (5'-ATTCTAGAGGCCGAGGCGGCCGACATG-d(T)30VN-3') and 1stST_STprimer

(5'-AAGCAGTGGTATCAACGCAGAGTGGCCATTATGGCCGGG-3') where the last three G were RNA bases. RNA was removed after the reaction using RNAse H (New England Biolabs) and the obtained cDNA was used as a template for long-distance PCR (LD-PCR) using primers LDPCR_5primer (5'-TTCCACCCAAGCAGTGGTATCAACGCAGAGTGG-3') and LDPCR_3primer (5'-GTATCGATGCCCACCCTCTAGAGGCCGAGGCGGCCGACA-3') and Phusion DNA polymerase (New England Biolabs) with the HF buffer. The associated PCR program consisted of a denaturation step of 30 seconds at 98°C followed by 22 cycles at 98°C (10 seconds), 55°C (15 seconds) and 72°C (4 mins) and a final elongation step of 5 mins at 72°C. Amplified DNA was then purified by chromatography using Sephacryl S500-HR columns (GE Healthcare) in order to optimize the size range of purified DNA.

Y187 yeasts (MATα, ura3-52, his3-200, ade2-101, trp1-901, leu2-3, 112, gal4Δ, met–, gal80Δ, MEL1, URA3::GAL1UAS-GAL1TATA-lacZ) were transformed using 2 μg of purified cDNA library (Thymus or feather) and 3 μg of SmaI-linearized pGAD-C1-rec using the lithium acetate method [44]. A total of one and two million clones were obtained with feather and Thymus cDNA respectively. Yeast libraries were aliquoted and conserved at -80°C as glycerol stocks.

### Yeast two hybrid screen

AH109 yeasts (MATa, trp1-901, leu2-3, 112, ura3-52, his3-200, gal4Δ, gal80Δ, LYS2::GAL1UAS-GAL1TATA-HIS3, MEL1, GAL2UAS-GAL2TATA-ADE2, URA3::MEL1UAS-MEL1TATA-lacZ) were transformed with pGBKT7-UL47 using lithium acetate yielding AH109-pGBKT7-UL47. Screening was performed by mating Y187 yeasts (MATα) containing either the thymus or the feather libraries with AH109-pGBKT7-UL47 (MATa). Briefly, a colony of AH109-pGBKT7-UL47 was seeded in Trp-depleted SD medium (Takara) and grown overnight. $5.10^8$ cells were then mixed with 1 mL of the cDNA library-containing Y187 yeasts and YPD (Yeast Extract/Peptone/Dextrone completed with adenine sulfate) medium to a final volume of 50 mL and placed on an orbital shaker at 55 rpm at 30°C. 24h later, cells were pelleted and suspended in 10 mL of –Trp – His –Leu (-WHL) SD medium and integrally plated on -WHL SD agar plates or on –Trp –Leu (-WL) plates for numbering of clones. Colonies containing interacting pairs were thus selected based on the transactivation of the HIS3 gene by the hybrid between BD-UL47 and the AD-associated interactor conferring the ability to yeasts to grow on histidine-depleted plates. The total number of clones was estimated to 1,800 and 2,100 for the feather and the thymus screens respectively. All clones were transferred onto –WHL plates containing 25 mM aminotriazole, an inhibitor of the chain of synthesis of histidine, to eliminate false positives. Interactions were further tested by the transactivation of the LacZ reporter gene (encoding β-galactosidase) by using the X-Gal assay on plates, yielding a number of clones positive for β-galactosidase of 120 and 1,900 for the feather and the thymus screens respectively. To narrow the number of clones to be tested and to limit redundancy, the size of the cDNA contained in every clone was assessed by colony PCR. Clones were then grouped according to the insert size. To recover the cDNA, yeasts were selected on –L plates until the pGBKT7-UL47 plasmid was lost and the pGAD-C1-rec-cDNA plasmid was then extracted from yeasts. An additional control for specificity was carried out by inserting selected cDNAs back into Y187 yeasts containing either pGBKT7-UL47 (positive control) or pGBKT7-UL13 or the empty pGBKT7 plasmid as negative controls and assessing for interaction through the β-galactosidase activity. At the end of the process, only one cDNA per screen encoded a specific interactor of pUL47. Sanger sequencing followed by BLAST alignment revealed that the two resulting cDNAs were distinct but encoded the same protein: *gallus gallus* p32/C1QBP/ gC1qR (NCBI ID: XM_025142057).

Liquid β-galactosidase quantification, β-galactosidase readout on plates and DNA extraction from yeasts were all performed as previously described [45].

### Plasmids

**Y2H plasmids.** The pGAD-C1-rec vector was made by inserting the sequence CACCCAAGCAGTGGTATCAACGCAGAGTGGCCATTATGGCCCGGGAAAAAACATGTCGGCCGCCTCGGCCTCTAGAGGGTGGGCATCGATACGG between the EcoRI and BamHI sites of pGAD-C1 [46]. pGBKT7-UL47 was obtained by amplifying the UL47 open reading frame (ORF) from BAC pRB-1B using Q5 High-Fidelity Polymerase (New England Biolabs) and cloning it into plasmid pGBKT7

using restriction sites EcoRI and BamHI (primer sequence in Table 1). pGBKT7-UL47 allows yeasts lacking the trp gene to grow on trp-depleted plates and fuses the GAL4 DNA-binding domain (BD) to the N-terminus of pUL47. pGBKT7-UL13 was constructed similarly using the same restriction sites. The relative toxicity in yeast of the two constructs was evaluated by measuring the average size of 65 colonies per construct and comparing it to the size of colonies bearing the pGBKT7 plasmid. This showed that the size of colonies of pGBKT7-UL47-transformed yeasts was reduced by 23% whereas it was unchanged in pGBKT7-UL13-transformed yeasts, indicating a mild toxicity of the UL47 construct.

pGBKT7-UL47 (SB1) was obtained by amplifying the UL47 ORF from purified genomes of the SB1 strain using Q5 High-Fidelity Polymerase (New England Biolabs) and cloning it into plasmid pGBKT7 using restriction sites EcoRI and BamHI (primer sequence in Table 1). pGBKT7-UL47 (HVT) was similarly obtained using BAC HVT TK_GFP [47] as a template and restriction sites SmaI and NotI.

All fragments of UL47 were cloned into pGBKT7 using specific primers as listed in Table 1.

All point mutations (aka SR mutants) were introduced into UL47 using a two-step PCR method. Briefly, a first fragment encompassing the start of UL47 up to the sequence containing the codon to be mutated was amplified using primers UL47lex_EcoRIf and SRxr (see Table 1) where x corresponds to the targeted SR mutation. A second fragment encompassing the sequence containing the codon to be mutated up to the end of the UL47 gene was amplified using primers SRxf and UL47lex_BamHIr. PCR fragments thus obtained were purified and used in equimolar quantities for a fusion PCR using primers UL47lex_EcoRIf and UL47lex_BamHIr. Mutant UL47Δ71-185 was similarly obtained using primers UL47d71-185_delF and UL47d71-185_delr instead of the respective SRx primers. All mutants were cloned into pGBKT7. All mutations were verified by Sanger sequencing.

**Avian cells expression.** The pcDNA3-V5-UL47 plasmid was described earlier [7]. The pcDNA3-myc-UL47 plasmid encoding full length pUL47 fused with a 6xMyc tag at its N-terminus was obtained similarly. pcDNA3-V5-UL47Δ71-185 and pcDNA3-myc-UL47Δ71-185 were obtained by amplifying UL47Δ71-185 from pGBKT7-UL47Δ71-185 and cloning it into pcDNA3-V5 and pcDNA3-myc using primers listed in Table 1. pcDNA3-V5 UL47 (SB1) was obtained by amplifying the UL47 ORF from purified genomes of the SB1 strain using Q5 High-Fidelity Polymerase (New England Biolabs) and cloning it into plasmid pcDNA3-V5 using restriction sites EcoRI and NotI. pcDNA3-V5-UL47 (HVT) was similarly obtained using BAC HVT TK_GFP as a template and restriction sites EcoRV and NotI.

Chicken p32 open reading frame was amplified from clone M9 obtained in the yeast two-hybrid screen and cloned into pcDNA3-V5, pcDNA3-myc and pEGFP-C1 yielding pcDNA3-V5-p32(M9), pcDNA3-myc-p32(M9) and pEGFP-C1-p32(M9) encoding the M9 fragment of p32 with a N-terminal V5, 6xMyc or EGFP tag respectively.

The Us3 ORF was PCR-amplified using Q5 High-Fidelity DNA polymerase (NEB) and pRB-1B as template and cloned into the EcoRI and XhoI restriction sites of pcDNA3-V5, yielding pCDN3-V5-Us3 encoding pUs3 with a N-terminal V5 tag. The UL32 and UL46 ORF were cloned similarly into pcDNA3-V5 and pcDNA3-myc using the HindIII and BamHI sites. The resulting plasmids encoded a C-terminal fusion of V5 or 6xMyc tags to pUL32 and pUL46. All transfection experiments (except for recovery of infectious virus from BACs) were carried out in ESCDL-1 cells.

## BACs

All recombinant Bacterial Artificial Chromosomes (BACs) used in this study were derived from the repaired rRB-1B BAC which contains the genome of the very virulent strain RB-1B of MDV as previously described [6,8]. The deletion of the sequence coding for residues 71-185 of pUL47 was introduced into the BAC using the two-step Red recombination technology [48], primers UL47d71-185_delF (GCCTATGATGACGATACATACGATACGCTAGAGGAGAGCGAGGA-TAATAGGATGACGACGATAAGTAGGG) and UL47d71-185_delR (CTAACACGGCTTGATCGTGATGCATGATTATAT-TCTCCTTCTCTAGACGAATTATCCTCGCTCTCCTCTAGCGTATCGTATGTATCGTCATCATAGGCACAACCAATTAA

**Table 1. List of oligonucleotides used in this study except for those specific to yeast-two hybrid (see Material & Methods).**

| Primer name | Role | Sequence | Template | Parental BAC |
|---|---|---|---|---|
| UL47d71-185_delF | Construction of rUL47Δ71-185 | GCCTATGATGACGATACATACGATACGCTAGAG-GAGAGCGAGGATAATAGGATGACGACGATAAG-TAGGG | pEP-KAN-S | rRB-1B |
| UL47d71-185_delR | Construction of rUL47Δ71-185 | CTAACACGGCTTGATCGTGATGCATGAT-TATATTCTCCTTCTCTAGACGAATTATCCTC-GCTCTCCTCTAGCGTATCGTATGTATCGTCAT-CATAGGCACAACCAATTAACCAATTCTGATTAG | pEP-KAN-S | rRB-1B |
| vUL47_GFPC-f | Construction of vUL47-GFP and vUL47Δ71-185-GFP | tcgagaaaagagccgtcgaaaccgcccgccgtgagccacaacgg-gcgaatATGGTGAGCAAGGGCGAGGAG | pEP-EGFP-in | rRB-1B/ rUL47Δ71-185 |
| vUL47_GFPC-r | Construction of vUL47-GFP and vUL47Δ71-185-GFP | tgccattattctacatccggagtaaaagtcccgccctcttccctacgt-caCTTGTACAGCTCGTCCATGCCG | pEP-EGFP-in | rRB-1B/ rUL47Δ71-185 |
| UL47lex_EcoRIf | Cloning of RB1B UL47 into pGBKT7 | CGAGAATTCCAAATGCCTTCTATGCATCGG | rRB-1B | |
| UL47lex_BamHIr | Cloning of RB1B UL47 into pGBKT7 | TTCGGATCCTCAATTCGCCCGTTGTGGCTC | rRB-1B | |
| UL47_SB1_EcoRIf | Cloning of SB1 UL47 into pGBKT7 and pcDNA3-V5 | AGGGAATTCCTGGCTCTACACCGTCATGG | SB1 purified genome | |
| UL47_SB1_BamHIr | Cloning of SB1 UL47 into pGBKT7 | GCTGGATCCTTACCGGTCCGTTTTTTTGAAG | SB1 purified genome | |
| UL47_SB1_NotIr | Cloning of SB1 UL47 into pcDNA3-V5 | GCGGCGGCCGCTTACCGGTCCGTTTTTTTGAAG | SB1 purified genome | |
| UL47_HVT_SmaI+1f | Cloning of HVT UL47 into pGBKT7 | AACCCCGGGGTCCTCTAAGTTCCGTTACG | rHVT BAC | |
| UL47_HVT_NotIr | Cloning of HVT UL47 into pGBKT7 and pcDNA3-V5 | TCGGCGGCCGCCTATTTAAAAGGGCGCGGTTC | rHVT BAC | |
| UL47_HVT_EcoRV+2f | Cloning of HVT UL47 into pcDNA3-V5 | CAAGATATCTGTCCTCTAAGTTCCGTTACG | rHVT BAC | |
| UL47_EcoRI1+1f | Cloning of RB1B UL47 into pcDNA3-myc | CCGGAATTCGCAAATGCCTTCTATGCATCGG-TATG | rRB-1B | |
| UL47_EcoRI1noATf | Cloning of RB1B UL47 into pcDNA3-V5 | CGAGAATTCCAAATGCCTTCTATGCATCGGTATG | rRB-1B | |
| UL47_NotIr | Cloning of RB1B UL47 into pcDNA3-V5 & myc | TCTGCGGCCGCTCAATTCGCCCGTTGTGGCT-CACG | rRB-1B | |
| UL47p309_BamR | Cloning of UL47 fragments ending with codon 309 | GGCGGATCCCGGACTCGTACCTTTACTATCAC | pGBKT7-UL47 | |
| UL47p309_EcoF | Cloning of UL47 fragments starting with codon 309 | GGTGAATTCCCGGCTTTGGCCTCCTTTCTCGAG | pGBKT7-UL47 | |
| UL47a579_BamR | Cloning of UL47 fragments ending with codon 579 | TCTGGATCCGGCAGTTTTCAGCCATATGTATTC | pGBKT7-UL47 | |
| UL47a579_EcoF | Cloning of UL47 fragments starting with codon 579 | CTGGAATTCGCCAGCGAGAGAAGTAAAAAG | pGBKT7-UL47 | |
| UL47g185_BamR | Cloning of UL47 fragments ending with codon 185 | CTCTGGATCCTCCTAGACTAGATTGG | pGBKT7-UL47 | |
| UL47g185_EcoF | Cloning of UL47 fragments starting with codon 185 | TAGGAATTCATCGTCTAGAGAAGGAG | pGBKT7-UL47 | |
| UL47g71_BamHIr | Cloning of RB1B UL47 1–71 into pGBKT7 | TTTGGATCCCCCATTATCCTCGCTCTCC | pGBKT7-UL47 | |
| UL47g71_EcoRIf | Cloning of UL47 fragments starting with codon 71 | GATGAATTCTTTGTAAAAACCATTCCTAATG | pGBKT7-UL47 | |
| UL47g119_BamHIr | Cloning of RB1B UL47 71-119 into pGBKT7 | TGTGGATCCACCGCGACGACCCGAAGG | pGBKT7-UL47 | |
| UL47g119_EcoRIf | Cloning of RB1B UL47 119–185 into pGBKT7 | TCGGAATTCTAGTGATACATCTCGTGATATG | pGBKT7-UL47 | |
| SR1f | Introduction of mutation SR1-AA into UL47 | ACAATTTGACAATGCAGCGGGGCGGGACCGTA-CACGATCTGG | pGBKT7-UL47 | |

*(Continued)*

**Table 1.** (Continued)

| Primer name | Role | Sequence | Template | Parental BAC |
|---|---|---|---|---|
| SR1r | Introduction of mutation SR1-AA into UL47 | CCAGATCGTGTACGGTCCCGCCCCGCTG-CATTGTCAAATTGT | pGBKT7-UL47 | |
| SR2f | Introduction of mutation SR2-AA into UL47 | GGTAGTGATACAGCTGCTGATATGATTAATGCATC | pGBKT7-UL47 | |
| SR2r | Introduction of mutation SR2-AA into UL47 | GATGCATTAATCATATCAGCAGCTGTATCACTACC | pGBKT7-UL47 | |
| SR3f | Introduction of mutation SR3-AA into UL47 | GCATCATTAAAAGCGGCAGCGAGGTCTAG | pGBKT7-UL47 | |
| SR3r | Introduction of mutation SR3-AA into UL47 | CTAGACCTCGCTGCCGCTTTTAATGATGC | pGBKT7-UL47 | |
| SR4f | Introduction of mutation SR4-AA into UL47 | TCGAGAGCGAGGGCTGCAAGACGCTCGTCATC | pGBKT7-UL47 | |
| SR4r | Introduction of mutation SR4-AA into UL47 | GATGACGAGCGTCTTGCAGCCCTCGCTCTCGA | pGBKT7-UL47 | |
| SR4.5f | Introduction of mutation SR5-AA into UL47 | GAAGACGCTCGTCAGCTGCACGACGCCATCG-TAATGC | pGBKT7-UL47 | |
| SR4.5r | Introduction of mutation SR5-AA into UL47 | GCATTACGATGGCGTCGTGCAGCTGACGAGC-GTCTTC | pGBKT7-UL47 | |
| SR5f | Introduction of mutation SR6-AA into UL47 | GCATTTTCGTGGGGGTGCCGCAGCAGCAG-CAACGGGTTCTC | pGBKT7-UL47 | |
| SR5r | Introduction of mutation SR6-AA into UL47 | GAACCCGTTGCAGATCGTGCGGCACCCCCAC-GAAAATGC | pGBKT7-UL47 | |
| M9_p32_EcoRIf | Cloning of chicken p32 (clone M9) into pcDNA3-V5 | ATCGAATTCGAGTGGGACAAAGCGTTCGCGCAG | pGAD-p32(clone M9) | |
| M9_p32_EcoRI+1f | Cloning of chicken p32 (clone M9) into pcDNA3-myc & pEGFP-C1 | TATGAATTCAGAGTGGGACAAAGCGTTCGCG-CAG | pGAD-p32(clone M9) | |
| UL32_HindIIIf | Cloning of RB1B UL32 into pcDNA3-V5 and myc | TCAAAGCTTATGGCCAACCGCCCTACAGAG | rRB-1B | |
| UL32_BamHIr | Cloning of RB1B UL32 into pcDNA3-V5 and myc | AATGGATCCCACGTAGACTCCTAATGTATGCTCG | rRB-1B | |
| UL46_HindIIIf | Cloning of RB1B UL46 into pcDNA3-V5 and myc | TGGAAGCTTATGAAGCGGCTCAGCTCTTCTG | rRB-1B | |
| UL46_BamHIr | Cloning of RB1B UL46 into pcDNA3-V5 and myc | ATAGGATCCATCGGTAGCCACCCTCAACCTAC | rRB-1B | |
| UL13_EcoRI0f | Cloning of RB1B UL13 into pGBKT7 | GACGAATTCGATACTGAATCAAAAAA-CAAAAAAACG | rRB-1B | |
| UL13_BamHIr | Cloning of RB1B UL13 into pGBKT7 | ACTGGATCCCTAGTTCCATAACAACAAATCAG | rRB-1B | |
| Us3_EcoRI0f | Cloning of RB1B Us3 into pCDNA3-V5 | GACGAATTCTCTTCGAGTCCGGAGGCAGAAAC-GATG | rRB-1B | |
| Us3_XhoIr | Cloning of RB1B Us3 into pCDNA3-V5 | GACCTCGAGTTACATATGAGCGGCAGTTATCG | rRB-1B | |
| UL47RT_f | Detection of UL47 transcripts by RT-PCR | GGAGAACACTTGTAGAAGCG | | |
| UL47RT_r | Detection of UL47 transcripts by RT-PCR | GAATGTCATAATTAATCGGCG | | |
| gC-RTPCRf2 | Detection of all UL44 transcripts by RT-PCR | GAGCTACGTTGGTTTCTACAATAAC | | |
| gC-RTPCRrev2 | Detection of all UL44 transcripts by RT-PCR | CATAGGGCAGTCATGATTATCC | | |
| UL18-RTf | Detection of UL18 transcripts by RT-PCR | CAAATGCCCCCTCCTACCAG | | |
| UL18qPCRr | Detection of UL18 transcripts by RT-PCR | CGCTTTTATATTGGCAGGGC | | |

CCAATTCTGATTAG) and plasmid pEP-Kan-S as template, yielding rUL47Δ71-185. The presence of the mutation was verified by PCR, by sequencing (NGS and Sanger) and by restriction fragment length polymorphism (RFLP) analysis.

Addition of GFP to the C-terminus of pUL47 was done using the same method using primers vUL47_GFPC-f (tcgagaaaagagccgtcgaaaccgcccgccgtgagccacaacgggcgaatATGGTGAGCAAGGGCGAGGAG) and vUL47_GFPC-r (tgccattattctacatccggagtaaaagtcccgccctcttccctacgtcaCTTGTACAGCTCGTCCATGCCG) as described previously [32] and plasmid pEP-EGFP-in as template (a kind gift from B. Kaufer, Freie Universität Berlin), and BACs rRB-1B and rUL47Δ71-185, yielding rUL47-GFP and rUL47Δ71-185-GFP respectively.

## Alignments

The sequences of human (NCBI Accession number NM_001212.4) and chicken (XM_040687101.2) p32/C1QBP were used for alignment using CLUSTALW software [49] which calculated an identity of 70% between the two sequences.

## Restriction fragment length polymorphism analysis (RFLP)

RFLP was carried out as described in previously [50]. Briefly, 1 µg of BAC DNA was digested with 10 units of EcoRI or XhoI (New England Biolabs) for 1h30 at 37 °C. DNA was then loaded onto a 0.6% agarose gel in Tris-Borate-EDTA (TBE) buffer preloaded with ethidium bromide and electrophoresis was carried out for five hours at 50V (35 mA). The profiles obtained were compared to the theoretical profiles generated using pDRAW32 software (v1.1.151 for Windows, Aca-clone software) with the following parameters: gel = 0.5%, minimum size cut-off=500 bp, DNA MW-marker = SmartLadder (Eurogentec).

## Full BAC sequencing (Illumina and ONT)

Illumina sequencing was outsourced to Novogene (Cambridge, UK) using one microgram of BAC DNA and Illumina Nova-seq sequencer.

MinION sequencing (Oxford Nanopore Technologies-ONT) was done in-house as follows. MinION DNA library was prepared using a Rapid Barcoding kit (SQK-RBK110 96) from 100 ng of BAC DNA, and the sequencing run was performed on a R9.4 Flongle (FLO-FLG001). Reads were basecalled and demultiplexed using Guppy v6.0.1 (Oxford Nanopore Technologies). Demultiplexed reads with an average quality higher or equal to 10 (phred score) and a minimum size of 500 bp were selected with nanofilt v.2.5.0, resulting in 5278 reads (mean read length 6964 nt) which were mapped on the RB-1B reference sequence (accession EF523390) using minimap2 v2.24-r1122 [51] with default parameters. Mutations found from the Illumina sequencing were visually checked on MinION reads using Interactive Genome Viewer v2.13.1 [52].

## Cells and viruses

Primary chicken embryonic skin cells (CESCs) were obtained from 12 days old specific pathogen-free (SPF) White Leghorn embryos as described previously [53] and maintained in William's modified E medium containing 1.5% chicken serum and 1% fetal calf serum. These cultures contain mainly dermal fibroblasts and some myoblasts.

ESCDL-1 cells are derived from chicken embryonic stem cells and have been characterized and previously validated for MDV replication [54]. These cells, which are fully permissive to MDV, can be efficiently transfected by standard methods such as lipofection, as opposed to CESCs. $10^6$ cells were transfected using 2 µl of JetOptimus (PolyPlus) and 0.5 µg of each plasmid following the manufacturer's instructions.

vUL47Δ71-185, vUL47Δ71-185-GFP and vUL47-GFP were obtained after transfection of $5.10^6$ CESCs with 4µg of rUL47Δ71-185, vUL47Δ71-185-GFP or rUL47-GFP respectively by electroporation using the Amaxa Nucleofector apparatus (program F024) with the Basic Nucleofactor Kit for Primary Mammalian Fibroblasts (Lonza). After four days of incubation, viral plaques could be observed. Viral stocks were obtained after a first passage on CESCs and the virus was not passaged more than three times and only on CESCs.

## Antibodies

The following primary monoclonal antibodies were used: anti-gB MDV (clone K11) [6], anti-VP22 MDV (clone B17) [55] anti-ICP4 MDV (clone E21) [6], anti-VP5 (clone F19) [56], anti-V5 (#46-0705, Invitrogen), mouse anti-myc (clone 9E10, Sigma) and rabbit anti-myc (C3956, Sigma). The rabbit anti-p32/C1QBP polyclonal antibody was obtained from Protein-tech (#24474-1-AP) and was raised against human p32.

## Co-immunoprecipitation assays

Transfected ESCDL1 cells were lysed 24h after transfection with cytoplasmic protein extraction buffer (CPEB; 50mM Tris-HCl pH8, 150mM NaCl, 5mM EDTA, 0,5% IGEPAL [(octylphenoxyl)polyethoxyethano] 630 (Sigma) and protease inhibitors (Complete, EDTA-free; Roche) for 30min on ice. In experiments involving nuclease, EDTA was omitted from CPEB and 90U of micrococcal nuclease (Sigma) was added or not. Cell lysates were incubated 30 mins at 37°C instead of on ice. Nuclei and cell debris were pelleted at 6000g for 5min. The supernatant was retained (cytoplasmic fraction) and the pellet was resuspended in nuclear protein extraction buffer (20mM HEPES KOH pH7.9, 420mM KCl, 20% Glycerol, 1mM EDTA, 2mM β-mercaptoethanol and protease inhibitors) for 30min on ice. The nuclear extracts were clarified at 13000g for 2min and the supernatants from the nuclear and cytoplasmic fractions were pooled to constitute the whole cell extract. All extracts were mixed with protein A for 30min at 4°C and centrifuged at 13000g for 2min to eliminate non-specific binding. A 30 μl sample of the supernatant, constituting 6% of the total extract, was removed before addition of the rabbit anti-myc antibody for 90min at 4°C. The immune complexes were collected on protein A-sepharose beads (Cytiva, CL-4B) by incubation for 1h at 4°C and washed three times in cold CPEB. 10 μL of Laemmli buffer (4X) were added to the pelleted beads which were stored at -20°C until analysis by Western-blot.

## Purification and analysis of RNA from cell lysates

Cells were lysed with CPEB as described above and a 100 μL sample was taken after the 30 minutes lysis period. RNA was purified from this sample using the Nucleospin RNA clean-up kit (MN) and following the manufacturer's instructions.

## Western blots

15 μL of sample were loaded onto a 10% SDS-PAGE. Proteins were transferred onto Porablot nitrocellulose membranes (Macherey-Nagel) which were probed with mouse primary antibodies against V5 or myc tags. Horseradish peroxidase-conjugated goat anti-mouse antibody (Sigma) was used for secondary detection in combination with the Immobilon Western HRP substrate (Millipore). Quantification of bands in each lane of the blot was done using the lane densitometric quantification function from Fiji/ImageJ software (v1.54j).

## Immunofluorescence

Mitochondria were labelled by adding MitoTracker CMTMRos Orange (Molecular Probes) to serum-free medium at a final concentration of 100nM for 30 minutes before fixation. Cells were washed twice with PBS and fixed with 4% paraformaldehyde at room temperature. Fifteen minutes later, cells were washed once with PBS and incubated 15 minutes at room temperature with 0.1% Triton X-100 diluted in PBS supplemented with 1% goat serum for blocking nonspecific antigenic sites. After an additional wash with PBS, cells were incubated with primary antibodies diluted 1:1000 in PBS at room temperature. One hour later, cells were washed three times with PBS and incubated with secondary antibodies diluted in PBS. Secondary antibodies used were AlexaFluor 488, 555 or 594-conjugated goat anti-mouse and AlexaFluor 488, 555 or 594-conjugated goat anti-rabbit antibodies (Molecular Probes). Incubation was done as for primary antibodies and cells were washed three times with PBS. Cells were then mounted on slides using a mixture of DABCO-Mowiol supplemented with 5μg/mL Hoechst (Molecular Probes). Images were obtained on an Axiovert 200 M inverted microscope (Zeiss) using

a 63x Plan-Apochromat objective (NA = 1.4, Zeiss), mounted with a Moment CMOS camera (Teledyne Photometrics) and driven by Inscoper hardware and software (v8.5.10). Images were processed using Fiji/ImageJ software (1.54j).

## FLIP experiments

Fluorescence Loss In Photobleaching experiments were carried out on CESCs that were infected for four days and maintained at 37 °C in a humidified chamber with 5% $CO_2$ during live-cell imaging. Imaging was conducted using an AX R confocal microscope (Nikon, Europe BV) equipped with a water-immersion 60x/1.2 objective, galvanometric scanner and a PFS module (Perfect Focus System, Nikon) to ensure focus is maintained throughout the entire experiment. A semi-automated protocol was performed using JOBS module. Briefly, a perinuclear region of the cell (4µm diameter circle) was manually selected and repeatedly photobleached every 10 seconds for a total duration of 10 minutes. A 488nm laser was used for bleaching (50% power and 2µs of dwell time). Fluorescence intensity of bleached- and unbleached-cells was imaged at regular intervals before every bleaching event. Image analysis was carried out using Fiji software (version 1.54p, [57]) by manually quantifying the nuclear mean fluorescence intensity of cells.

## Plaque size assay

CESCs seeded in 6-well plates were infected with 100 pfu of vRB-1B or vUL47Δ71-185 and incubated in medium containing 1% carboxy-methylcellulose to prevent the formation of secondary plaques. Cells were fixed four days later with 4% paraformaldehyde and permeabilized with 0.1% Triton X-100. They were then blocked using 1% of bovine serum albumin (Fraction V, GE Healthcare) for 20 minutes at room temperature. Plaques were labeled using a mix of monoclonal antibodies directed against gB, ICP4 and VP22 and AlexaFluor 488-conjugated goat anti-mouse secondary antibody (Molecular Probes). Images of at least 50 plaques per condition were obtained on an Axiovert 200 M inverted microscope (Zeiss) using a 5x Fluar long-distance objective (NA = 0.25). The size of plaques was determined using Fiji software (v1.54j).

## DNA isolation and quantification of viral genomes by TaqMan qPCR

Venous occipital sinus blood samples were collected into tubes containing 3% sodium citrate. Peripheral blood mononuclear cells (PBMC) were isolated using density gradient centrifugation on MSL (Eurobio, France). The DNA extractions were performed using the DNA Purification "Blood or Body Fluids Spin" protocol of the QIAamp DNA Mini Kit (Qiagen, Germany). Incubation time with proteinase K at 56 °C was extended from 10 min to 2 h to increase DNA yield.

The feather tip of approximatively 3–4 growing feathers was manually collected and DNA was extracted using the "tissues protocol" of the QIAamp DNA Mini Kit (Qiagen). Samples were incubated overnight at 56 °C with proteinase K to ensure efficient lysis. DNA was extracted from 10 mg of dust accumulated in the isolation unit for one week using the same protocol as for feather tips. DNA concentrations were measured with a NanoDrop spectrophotometer.

Real-time PCR was performed using TaqMan technology, as previously described [58,59]. Both iNos and the ICP4 probes were tagged with FAM-BHQ1. All qPCR were performed independently with 250 ng DNA, ten pmol of each gene-specific primer, five pmol of the gene-specific probe in a 20 µL volume on a CFX96 Real Time C1000 Touch Thermal Cycler (BioRad, Marnes-la-Coquette, France). The results were analyzed using the CFX Manager software (version 3.1, BioRad). For each sample, viral DNA (based on ICP4 gene) and cellular DNA (based on iNos) were quantified independently in triplicates. The positive cutoff point corresponded to 10 copies. For each sample, the number of MDV genome copies was calculated per $10^6$ cells.

## RNA isolation and RT-PCR

RNA was extracted from skin or feather tips using 2–3 skin samples of approximatively 15 $mm^2$ containing feather follicles and the feather tips of 2–3 large feathers were collected in 2 mL Eppendorf tubes containing a mix of (1:5) or (5:1)

respectively of 1.4mm diameter and 2.8mm diameter ceramic beads. One milliliter of Trizol LS (Sigma) was added to the mixture and the samples were ground with manual shaking until the solution was cloudy. Trizol was transferred into a new centrifuge tube and debris was removed by centrifugation. 200 µL of chloroform were added and mixed with Trizol. Phase separation was obtained after centrifugation at 12,000g for 15 minutes. The aqueous phase was recovered and RNA was precipitated by the addition of one milliliter of RNase-free isopropanol and centrifugation 12,000g for eight minutes. The RNA pellet was washed with RNase-free 70% ethanol, dried and recovered in 15 µL of RNase-free water. RNA was then incubated with 2U of DNAse I (Promega) in DNase I buffer and 40U of RNasin (Promega) for one hour at 37°C. DNase was inactivated by incubation at 60°C for ten minutes before proceeding to the RT-PCR.

Reverse-transcription was carried out using the iScript Reverse Transcription Supermix (BioRAD) and 1µg of DNase-treated RNA according to the manufacturer's protocol. PCRs were done on 0.5µL of cDNA in a volume of 25µL using the GoTaq Flexi G2 Taq DNA polymerase (Promega) following the manufacturer's instructions. All PCR reactions were done with the following program: 3 minutes of denaturation at 94°C and 30–35 cycles of denaturation at 94°C for 30 seconds, annealing at 52°C for 30 seconds and extension at 72°C for 40 seconds. A final extension of five minutes was added at the end of the program. For cDNA obtained from infected cells, 30 cycles of amplification were used whereas for cDNA obtained from tissue, 35 cycles of amplification were used.

### Animal experiments

Specific pathogen-free (SPF) and MDV maternal antibody-free White Leghorn chicks (B13/B13 haplotype) of two weeks of age were obtained from INRAE animal facilities. Animals were separated into two groups of seventeen animals housed in two independent isolation units. Nine animals were inoculated with the virus while eight other animals were left unin-fected ("contacts") to measure natural viral transmission. Animals were inoculated by intramuscular injection of 3000 pfu of vRB-1B or vUL47Δ71-185 in 200µL of William's E medium. The experiment was terminated 68 days post-inoculation.

### Statistics

A two-tailed Wilcoxon-Mann-Whitney test was used for FLIP and qPCR experiments with small samples (<30) or exper-iments where the sample distribution was not Gaussian as determined by a Shapiro-Wilk test for $p > 0.05$. All statistical analyses were performed using Prism6 software (GraphPad).

### Acknowledgments

We thank Benedikt Kaufer for the gift of the pEP-EGFP-in plasmid and Venugopal Nair for sharing the HVT TK-GFP BAC. We are grateful to Karine Praud for the preparation of DNA libraries for MinION sequencing and Ines Delhomme for her help in the construction of Y2H cDNA libraries. We thank Mélanie Chollot for the regular preparation of CESCs. All animal experiments were carried out at the Experimental Infectiology Facilities (PFIE) of INRAE (Nouzilly). We are grateful to Sébastien Lavillate, Arnaud Faurie, Mylène Girault and Olivier Dubes for their invaluable help with animal care and han-dling and to Thierry Chaumeil and Céline Barc for supervising these experiments.

### Author contributions

**Conceptualization:** David Pasdeloup.

**Data curation:** Sébastien Leclercq, David Pasdeloup.

**Formal analysis:** Sylvie Rémy, Sébastien Leclercq, Julien Pichon, David Pasdeloup.

**Funding acquisition:** Caroline Denesvre, David Pasdeloup.

**Investigation:** Mallorie Durand, Aurélien Chuard, Sylvie Rémy, Katia Courvoisier-Guyader, Sébastien Leclercq, Julien Pichon, David Pasdeloup.

**Methodology:** Mallorie Durand, Aurélien Chuard, Sébastien Leclercq, Julien Pichon, David Pasdeloup.

**Project administration:** David Pasdeloup.

**Resources:** Mallorie Durand, Aurélien Chuard, Katia Courvoisier-Guyader, Sébastien Leclercq, Julien Pichon, David Pasdeloup.

**Software:** Sébastien Leclercq, Julien Pichon.

**Supervision:** David Pasdeloup.

**Validation:** Mallorie Durand, Sébastien Leclercq, Caroline Denesvre, David Pasdeloup.

**Visualization:** Mallorie Durand, Caroline Denesvre, David Pasdeloup.

**Writing – original draft:** David Pasdeloup.

**Writing – review & editing:** Mallorie Durand, Aurélien Chuard, Sébastien Leclercq, Julien Pichon, Caroline Denesvre.

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
