## [Decision Letter · Decision Letter 0]

16 Apr 2025

The pUL47 tegument protein of Marek’s Disease Virus interacts with p32/C1QBP to promote horizontal transmission

PLOS Pathogens

Dear Dr. Pasdeloup,

Thank you for submitting your manuscript to PLOS Pathogens. After careful consideration, we feel that it has merit but does not fully meet PLOS Pathogens's publication criteria as it currently stands. Therefore, we invite you to submit a revised version of the manuscript that addresses the points raised during the review process.

Please submit your revised manuscript within 60 days Jun 15 2025 11:59PM. If you will need more time than this to complete your revisions, please reply to this message or contact the journal office at plospathogens@plos.org. Please include the following items when submitting your revised manuscript:

We look forward to receiving your revised manuscript.

Kind regards,

Ekaterina E. Heldwein

Academic Editor

PLOS Pathogens

Robert Kalejta

Section Editor

PLOS Pathogens

Editor-in-Chief

PLOS Pathogens

orcid.org/0000-0003-2946-9497

Editor-in-Chief

PLOS Pathogens

orcid.org/0000-0002-7699-2064

**Additional Editor Comments:**

In the revised version, please, make sure to address all the criticisms. In particular, Reviewer 1 was concerned about the lack of an alternative hypothesis to explain the data that contradict the initial hypothesis and experiments to validate it. Reviewer 2 questioned whether UL47 and p32 interact directly and the role of this interaction in infection.

**Journal Requirements:**

1) Please upload all main figures as separate Figure files in .tif or .eps format. For more information about how to convert and format your figure files please see our guidelines: 

2) Please ensure that the funders and grant numbers match between the Financial Disclosure field and the Funding Information tab in your submission form. Note that the funders must be provided in the same order in both places as well.

**Reviewers' Comments:**

Reviewer's Responses to Questions

**Part I - Summary**

Reviewer #1: In this report by Durand et al., the potential interaction of Marek’s disease virus (MDV) tegument protein UL47 with cellular proteins was investigated. The authors used a yeast two-hybrid system to identify cellular protein p32/C1QBP. The evidence for this interaction is strong, with interactions identified in two different chicken tissue libraries and extensive protein-protein interaction analyses, including co-immunoprecipitation and co-localization in immunofluorescence assays. The authors concluded that the interaction of UL47 and p32 was located within aa 71-185 of UL47. Based on one of the functions of p32 in regulating mRNA splicing and their previous studies investigating UL47’s role in the splicing of MDV UL44 (gC), they hypothesized that the interaction between p32 and UL47 was required for regulating UL44 splicing. However, they never tested whether this interaction affected UL44 splicing in vitro and continued with in vivo studies using a recombinant virus in which the proposed interacting residues on UL47 (71-185) were deleted. They found no difference in the localization of UL47 in cell culture and no difference in replication in experimentally infected chickens compared to wild-type MDV. However, the deletion mutant was unable to transmit to chickens, similar to their previous studies in which the complete UL47 gene was deleted. Analysis of UL44 mRNA splicing in feathers and skin revealed no differences between the MDV wild-type and the deletion mutant, disproving their hypothesis. The manuscript's strengths include the convincing interaction data, the sound in vivo experiments, and, particularly, the use of a natural virus-host in vivo model system to test their hypothesis. The significant weaknesses include the oversimplified interpretation of the results, and further characterization and experiments are needed to address their in vivo results.

Reviewer #2: Durand et al. characterize a distinct domain in the MDV tegument protein, UL47, which is essential for horizontal transmission in its native host (chickens). The data demonstrating an absence of horizontal transmission in their UL47 del71-185 mutant virus as compared to wild type is very convincing and poses an interesting model for decoupling replication and transmission. We would ask the authors to strengthen their conclusions regarding the interaction between UL47 and p32. In particular, there is an absence of data demonstrating the UL47-p32 interaction in the context of MDV infection. There is also no data directly connecting the lack of p32 interaction to the absence of horizontal transmission. We find the article is well written, although improvements could be made to the figures and legends for readability.

Reviewer #3: In this manuscript, the authors investigated how the pUL47 tegument protein contributes to Marek’s disease virus transmission. They used yeast-two hybrid screens to identify interaction partners of pUL47. They identified the cellular p32 (C1QBP) protein and mapped the interaction domain in pUL47. Deletion of this interaction domain did not affect other functions of pUL47 or pathogenesis of the respective mutant virus; however, virus shedding was completely abrogated in a gC independent manner. This study is well written and contains exciting data for the field and beyond, as the role of pUL47 in virus transmission can only be assess in very few natural virus/host models. The authors should address the following points prior to publication.

**Part II – Major Issues: Key Experiments Required for Acceptance**

Reviewer #1: 1. The overall subject of the paper involves the interaction between UL47 and p32, which is believed to be crucial for regulating mRNA splicing. However, their data completely refutes this hypothesis, which is not a problem per se, but, alternative mechanisms are not addressed. For example, the authors state that removing aa 71-185 does not affect UL47 localization in vitro and assume this does not affect its overall function. However, is a negative result. It has been published that localization of MDV UL47 is nuclear in cell culture and nucleocytoplasmic in skin cells in chickens. Similar results are seen for HSV UL47 in vitro. The authors should address the localization of UL47 (del 71-185) in skin cells to fully conclude that localization is not affected by this deletion. Alphaherpesvirus UL47 proteins are highly complex, and their localization in cells is tightly regulated, at least in the context of HSV studies. Thus, the authors cannot conclude localization has not been affected using in vitro studies, as the protein is not localized normally in cell culture. It is possible that this region is crucial for the shuttling of UL47 from the nucleus to the cytoplasm in vivo that would likely be important for its function.

2. In addition to the localization results, the removal of this region of UL47 is likely to affect multiple aspects of the protein's function. The results show that mRNA splicing was unaffected by the removal of these sequences; however, transmission was completely abrogated. What else is present in this region that could interfere with UL47 function in skin cells? Arginine rich regions can have many important functions that are not addressed here. Further studies, or at least hypotheses, should be developed and addressed. It would also be worthwhile to define the interaction of UL47 with p32 better and mutate small sequences rather than removing large portions of the protein that likely affect multiple aspects of UL47’s function.

3. The results in p32 localization and UL47 interactions is confusing. The localization of p32 (M9, D30 truncated and epitope-tagged) and cellular p32 is conflicting and has not been fully addressed. The results suggest that these do not identify the same protein. Figure 3D is not convincing. The full p32 cDNA should be cloned, and its localization and interaction with UL47 should be addressed. The current results are not well supported and raise more questions for this study.

Reviewer #2: 1. UL47 is an RNA binding protein, as demonstrated by prior studies with MDV and HSV1. This makes it probable that the interaction between UL47 and p32 is mediated by RNA. The authors should repeat their Co-IP experiments with benozonase (or a comparable nuclease) to evaluate if the binding is lost after nuclease treatment. The current data cannot conclude whether the UL47-p32 interaction is direct (protein-protein) or indirect (protein-RNA-protein).

2. Currently there is minimal data supporting UL47-p32 interaction or colocalization in the context of infection. Almost, if not all, of the UL47-p32 interaction findings were made via ectopic expression of labeled constructs. Only one figure panel (Fig. 3C) used an actual virus and this panel did not recapitulate the findings of their expression constructs (likely due to antibody limitations). The authors should provide additional data to support the existence of the UL47-p32 interaction in the context of infection. I’m curious if the nuclear speckles persist during an actual infection, or if these seed proper replication compartments. If possible, a time course examining UL47-p32 interactions during infection would be helpful in clarifying the functional outcome of this interaction.

3. We would ask the authors to strengthen their data supporting a lack of interaction between UL47 del71-185 and p32 (Fig. 5). In Fig. 5C the decrease in V5 levels after myc IP for the del71-185 mutant appears similar to the decrease in total UL47 expression for the mutant. I’m also uncertain which protein band is p32 in the V5 panel of Fig 5C (myc IP), as there are two bands present. This doublet was not present in prior p32 blot images, which leads me to question if it’s been cropped or is a new band. We would ask the authors to provide additional replicates (with more comparable expression of UL47 constructs) or an alternative method to demonstrate UL47 del 71-185 fails to interact with p32 (e.g. PLA immunofluorescence). This is particularly necessary as the IF in Fig. 5B demonstrates colocalization of UL47 del 71-185 with p32.

Reviewer #3: (No Response)

**Part III – Minor Issues: Editorial and Data Presentation Modifications**

Reviewer #1: 1) The authors utilize viral genomes as evidence of viral shedding, a standard method for analyzing MDV transmission. They found little difference in genomes shed but never addressed whether infectious virus is shed between the two viruses. Thus, it appears that the birds are shedding the virus; so, where is the defect in transmission? Further discussion of this should be explained.

Reviewer #2: 1. Remove the text “suggest that pUL47 may contribute to reprogramming cellular splicing to promote transmission” from the Abstract. This is speculative and best suited to the discussion.

2. Generally data presentation and transparency should be improved including the following:

a. Label western blots with size markers (Fig. 1C-D, 5C-D)

b. Clearly indicate the number of replicates for each panel, for n≤5 data points should be present in the graphs (Fig. 4C, 7)

c. In panel 6C how many plaques were measured & how many biological replicates did this correspond to.

3. Figure legends could have increased clarity including the following:

a. Statistical test used and what conditions were compared. The legend should detail what p-value range the asterisk denotes (Fig. 4C, 6C, 7C-D)

b. Indicate either the cell type or source material (animal vs. cell culture) assessed in the assay. For instance Fig. 2 , 3, 5 just says “Cells were transfected”. Based on the methods section I’m guessing they used CESCs, but it would be preferable if they said CESC instead of cells.

Reviewer #3: 1) The authors make the exciting observation that virus transmission is abrogate upon deletion of UL47 aa71-185 in vivo. Their data clearly shows that the UL47Δaa71-185 virus is still fully pathogenic, but that transmission is completely abrogated. However, they also show that shedding of this virus occurs efficiently, as high virus levels were detected in the dust. This suggests that either the shed UL47Δaa71-185 virus is defective or that it can not enter and/or infect the target cell in the lung. As addressing these questions is beyond the scope of this manuscript, the authors should at least extensively discuss these options in the Discussion section.

2) Viruses do not “survive” (e.g. line 56 “key part of viruses’ life cycle for survival in the host population.”). They spread and/or are maintained in a population. Please rephrase this.

3) Line 64ff: correct to “alphaherpesviruses”

4) Line 66: The authors state that the MDV “genome is integrated to the cellular genome during latency”. The virus specifically integrates into host telomeres. Please correct his and provide a reference for your statement.

5) Line 70: “excretion from the FFE”. Excretion has not been formally shown to my knowledge. Maybe use “release” instead.

6) The fond size in Table 1 is two small. Please optimize.

7) Figure 4B: please properly align the labels right and left of the UL47 overview (especially for 19-21)

8) Line 164ff: “Invalidate” does not fit in this context (e.g. “does not invalidate functions of pUL47”). Abrogate or block would fit better.

9) Figure 8B. Please indicate the full length, 104 and 145 on the right as nicely done for Figure 8A.

10) Line 282: please properly cite the Liu et al. paper.

PLOS authors have the option to publish the peer review history of their article (what does this mean? ). If published, this will include your full peer review and any attached files.

**Do you want your identity to be public for this peer review?** For information about this choice, including consent withdrawal, please see our Privacy Policy .

Reviewer #1: No

Reviewer #2: No

Reviewer #3: **Yes: ** Benedikt B. Kaufer

**Figure resubmission:**

**Reproducibility:**



---

## [Decision Letter · Decision Letter 1]

6 Aug 2025

Dear Dr Pasdeloup,

We are pleased to inform you that your manuscript 'The pUL47 tegument protein of Marek’s Disease Virus interacts with p32/C1QBP to promote horizontal transmission' has been provisionally accepted for publication in PLOS Pathogens.

Best regards,

Ekaterina E. Heldwein

Academic Editor

PLOS Pathogens

Robert Kalejta

Section Editor

PLOS Pathogens

Sumita Bhaduri-McIntosh

Editor-in-Chief

PLOS Pathogens

orcid.org/0000-0003-2946-9497

Michael Malim

Editor-in-Chief

PLOS Pathogens

orcid.org/0000-0002-7699-2064

Reviewer Comments (if any, and for reference):

Reviewer's Responses to Questions

**Part I - Summary**

Reviewer #1: Please see original review

Reviewer #2: Durand et al. characterize a distinct domain in the MDV tegument protein, UL47, which is essential for horizontal transmission in its native host (chickens). The data demonstrating an absence of horizontal transmission in their UL47 del71-185 mutant virus as compared to wild type is very convincing and poses an interesting model for decoupling replication and transmission. The authors provide data supporting a model in which the UL47 deletion has differential subcellular localization and support a model wherein this alters viral splicing to ultimately impact transmission.

Reviewer #3: In this manuscript, the authors investigated how the pUL47 tegument protein contributes to Marek’s disease virus transmission. They used yeast-two hybrid screens to identify interaction partners of pUL47. They identified the cellular p32 (C1QBP) protein and mapped the interaction domain in pUL47. Deletion of this interaction domain did not affect other functions of pUL47 or pathogenesis of the respective mutant virus; however, virus shedding was completely abrogated in a gC independent manner. This study is well written and contains exciting data for the field and beyond, as the role of pUL47 in virus transmission can only be assess in very few natural virus/host models. In this resubmission of the manuscript PPATHOGENS-D-25-00721R1, the authors addressed this reviewer's comments and substantially improved the manuscript, which is now suitable for publication.

**Part II – Major Issues: Key Experiments Required for Acceptance**

Reviewer #1: Please see original review

Reviewer #2: We find our prior concerns have been appropriately addressed in the revised submission and have no additional requests.

Reviewer #3: none

**Part III – Minor Issues: Editorial and Data Presentation Modifications**

Reviewer #1: Please see original review

Reviewer #2: (No Response)

Reviewer #3: none

PLOS authors have the option to publish the peer review history of their article (what does this mean? ). If published, this will include your full peer review and any attached files.

**Do you want your identity to be public for this peer review?** For information about this choice, including consent withdrawal, please see our Privacy Policy .

Reviewer #1: No

Reviewer #2: No

Reviewer #3: No

---

## [Editor Report · Acceptance letter]

Dear Dr Pasdeloup,

We are delighted to inform you that your manuscript, "The pUL47 tegument protein of Marek’s Disease Virus interacts with p32/C1QBP to promote horizontal transmission," has been formally accepted for publication in PLOS Pathogens.

Best regards,

Sumita Bhaduri-McIntosh

Editor-in-Chief

PLOS Pathogens

orcid.org/0000-0003-2946-9497

Michael Malim

Editor-in-Chief

PLOS Pathogens

orcid.org/0000-0002-7699-2064